# Structural and functional studies of the first tripartite protein complex at the *Trypanosoma brucei* flagellar pocket collar

Charlotte Isch[1‡], Paul Majneri[2‡], Nicolas Landrein[1‡], Yulia Pivovarova[2], Johannes Lesigang[2], Florian Lauruol[1], Derrick R. Robinson[1], Gang Dong[2]*, Mélanie Bonhivers[1]*

1 Univ. Bordeaux, CNRS, Microbiologie Fondamentale et Pathogénicité, UMR 5234, Bordeaux, France,
2 Max Perutz Labs, Vienna BioCenter, Medical University of Vienna, Vienna, Austria

‡ These authors are co-first authors on this work.
* gang.dong@meduniwien.ac.at (GD); melanie.bonhivers@u-bordeaux.fr (MB)

## Abstract

The flagellar pocket (FP) is the only endo- and exocytic organelle in most trypanosomes and, as such, is essential throughout the life cycle of the parasite. The neck of the FP is maintained enclosed around the flagellum *via* the flagellar pocket collar (FPC). The FPC is a macromolecular cytoskeletal structure and is essential for the formation of the FP and cytokinesis. FPC biogenesis and structure are poorly understood, mainly due to the lack of information on FPC composition. To date, only two FPC proteins, BILBO1 and FPC4, have been characterized. BILBO1 forms a molecular skeleton upon which other FPC proteins can, theoretically, dock onto. We previously identified FPC4 as the first BILBO1 interacting partner and demonstrated that its C-terminal domain interacts with the BILBO1 N-terminal domain (NTD). Here, we report by yeast two-hybrid, bioinformatics, functional and structural studies the characterization of a new FPC component and BILBO1 partner protein, BILBO2 (Tb927.6.3240). Further, we demonstrate that BILBO1 and BILBO2 share a homologous NTD and that both domains interact with FPC4. We have determined a 1.9 Å resolution crystal structure of the BILBO2 NTD in complex with the FPC4 BILBO1-binding domain. Together with mutational analyses, our studies reveal key residues for the function of the BILBO2 NTD and its interaction with FPC4 and evidenced a tripartite interaction between BILBO1, BILBO2, and FPC4. Our work sheds light on the first atomic structure of an FPC protein complex and represents a significant step in deciphering the FPC function in *Trypanosoma brucei* and other pathogenic kinetoplastids.

## Author summary

Trypanosomes belong to a group of zoonotic, protist, parasites that are found in Africa, Asia, South America, and Europe and are responsible for severe human and animal diseases. They all have a common structure called the flagellar pocket (FP). In most trypanosomes, all macromolecular exchanges between the trypanosome and the environment

**Data Availability Statement:** The numerical data used in Figs 3D and 4D are included in S1 Data. Coordinates and structure factors of the crystal structure of the BILBO2-NTD/FPC4-CTD complex

have been deposited in the Protein Data Bank (PDB) under accession code 7a1i. https://doi.org/10.2210/pdb7A1I/pdb.

**Funding:** This work was supported by the CNRS and the University of Bordeaux to DRR and MB, the LabEx ParaFrap [ANR-11-LABX-0024] to DRR, the Max Perutz Labs and grant [P24383-B21] and [I4960-B] from the Austrian Science Fund (FWF) to GD, the ANR-FWF PRCI [ANR-20-CE91-0003] to MB. YP was supported by the "Integrative Structural Biology" PhD program [W-1258 Doktoratskollegs] funded by the FWF and CI was supported by the LabEx Parafrap PhD program [ANR-11-LABX-0024]. The funders had no role in study design, data collection and analysis, decision to publish, or preparation of the manuscript.

**Competing interests:** The authors have declared that no competing interests exist.

occur *via* the FP. The FP is thus essential for cell viability and evading the host immune response. We have been studying the flagellar pocket collar (FPC), an enigmatic macromolecular structure at the neck of the FP, and demonstrated its essentiality for the biogenesis of the FP. We demonstrated that BILBO1 is an essential protein of the FPC that interacts with other proteins including a microtubule-binding protein FPC4.

Here we identify another *bona fide* FPC protein, BILBO2, so named because of close similarity with BILBO1 in protein localization and functional domains. We demonstrate that BILBO1 and BILBO2 share a common N-terminal domain involved in the interaction with FPC4, and illustrate a tripartite interaction between BILBO1, BILBO2, and FPC4. Our study also provides the first atomic view of two FPC components. These data represent an additional step in deciphering the FPC structure and function in *T. brucei*.

## Introduction

Trypanosomatids include many parasites of major medical and economic importance that cause several of the 20 World Health Organization's listed neglected tropical diseases. These flagellated parasites share several unique features: a single mitochondrion with its compact genome (the kinetoplast, K), a flagellar pocket (FP), and a microtubule-based cytoskeleton to maintain cell shape and flagellar motility that plays crucial roles in life and cell cycle [1]. The FP is an invagination of the plasma membrane enclosing the base of the flagellum. In most trypanosomes, endo- and exocytosis occur exclusively through the FP. It thus provides the sole surface for numerous important receptors making them inaccessible for components of the innate immune system of the host. Moreover, the FP is responsible for sorting all parasite surface glycoproteins targeted to, or recycling from, the pellicular membrane and for removal of host antibodies from the cell surface. As such, the FP is a key player in protein trafficking, cell signalling and immune evasion [2]. Because it is hidden from the cell surface and sequesters important receptors, the FP is an attractive drug target. However, it has not been exploited as such because structural components of this organelle are still poorly characterized.

During its life cycle, *T. brucei* is transmitted to the mammalian host *via* a blood meal of an infected tsetse fly. The parasite differentiates to several different forms in the insect and the mammalian host, among them the procyclic form (PCF) in the fly's midgut, and the bloodstream form (BSF) in the mammalian bloodstream. Organelle positioning and segregation during the cell and parasite cycle show a high degree of coordination and control [3].

The shape of the trypanosome cell is maintained by a microtubule-based corset and by a flagellum laterally attached along the cell body. The flagellum is involved in cell mobility, kinetoplast segregation, and signal transduction [1]. It extends from the mature basal body (BB, tethered to the kinetoplast), and exits the cell through the FP. It then runs along the length of the cell while remaining attached to the cell body *via* the flagellum attachment zone (FAZ). Four specialized microtubules (the microtubule quartet, MTQ) nucleate at the BBs and extend around the FP, insert into the microtubule corset, and run as part of the cytoplasmic portion of the FAZ as far as to the anterior end of the cell body. The bulb-like FP is maintained by a ring-like cytoskeletal structure, the flagellar pocket collar (FPC), which encircles the neck of the FP around the exit site of flagellum beneath the cell surface [4,5]. The FPC is a complex structure, and in addition to its attachment to the flagellum, it is also attached to the microtubule cytoskeleton. Overlapping with the FPC is the hook complex (HC), a cytoskeleton-associated structure that is superimposed on top of the FPC throughout the cell cycle. The MTQ threads between these two structures [6–8].

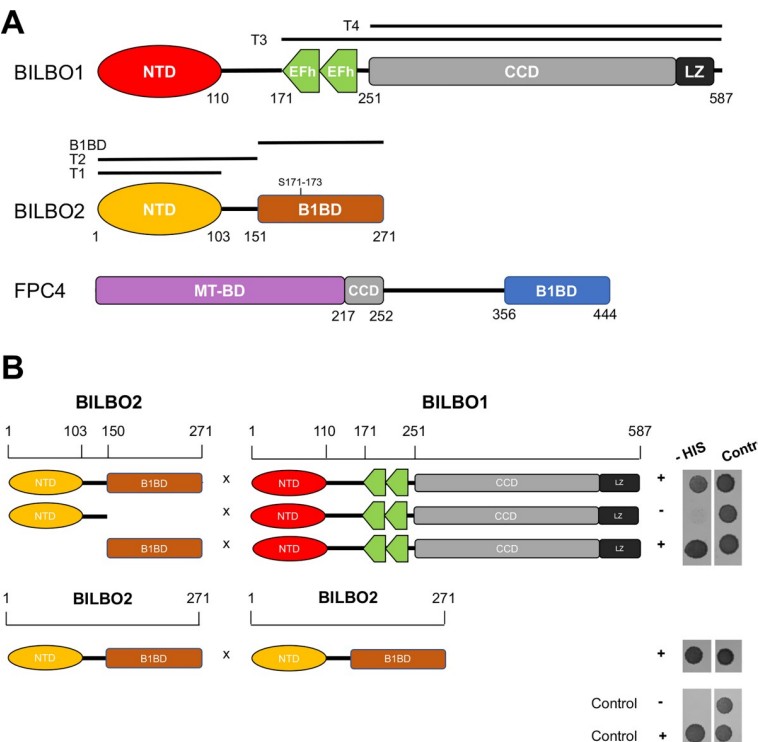

**Fig 1. Schematic representation of the BILBO1, FPC4, and BILBO2 secondary structures and Y2H interaction tests. A.** The BILBO1 domains T3 (aa 171–587) and T4 (aa 251–587) were previously described in [9]. BILBO2 is presented as three domains: T1 (aa 1–103), T2 (aa 1–150), and the BILBO1-binding domain (B1BD, aa 151–271). FPC4 is presented as three domains: the microtubule binding domain (MT-BD), the coiled-coil domain (CCD), and the BILBO1 binding domain (B1BD). **B.** Y2H assay with full-length or domains of BILBO2 as bait and BILBO1 as prey, and of BILBO2 as prey and bait tested on minus histidine medium (-HIS). Loading control was on medium plus histidine. The positive control involved the previously demonstrated interaction between the p53 and T-antigen proteins, whereas the negative control involved Lamin and T-antigen proteins that do not interact.

BILBO1 is the first identified FPC protein, with an indispensable role for the parasite [4]. RNA interference (RNAi) knockdown of BILBO1 in PCF cells prevents the biogenesis of a new FPC, a new FP, and a new FAZ, suggesting that the FPC is required for the biogenesis of numerous structures and their functions in the cell. In BILBO1 RNAi cells, the newly formed flagellum locates at the extended posterior end of the cell and is detached from the cell body. Furthermore, knockdown of BILBO1 is lethal in both PCF and BSF cells.

BILBO1 is a modular protein with four structural domains [9,10] (Fig 1A). The globular N-terminal domain (NTD) is followed by two calcium-binding EF-hand motifs (EFhs), a central coiled-coil domain (CCD), and a C-terminal leucine zipper (LZ). The LZ is necessary but not sufficient for FPC targeting of BILBO1. The CCD allows the formation of filaments by the formation of antiparallel dimers that can extend into a polymer by the interdimer interaction between adjacent LZs. Indeed, BILBO1 was shown to form micrometre-long polymers and helical structures *in vivo*, *in vitro* and in a mammalian cell environment [4,9–12].

The reported high-resolution NMR and crystal structures of the BILBO1-NTD demonstrate that it unexpectedly adopts a ubiquitin-like fold [13,14]. The C-terminal tail of the NTD is well-folded and rigidly wraps around the distal end of the elongated core structure. This tail helps to form a well-defined horseshoe-like pocket that harbours multiple highly conserved aromatic residues. On one side of the hydrophobic pocket, a gap is formed that leads to a pronounced negative trench at the bottom of the structure. Mutation of key residues within the

pocket affect cell viability and impair the BILBO1 function in trypanosomes. Further, abolishing the $Ca^{2+}$-binding ability of the EFh influences the shape and length of the polymers of BILBO1, disrupts the FPC structure, and affects trypanosome cell viability [9,10,15]. Deletion of both globular domains (*i.e.* NTD and EFh) leads to shorter polymers than those formed by the full-length BILBO1 [10], suggesting their role in facilitating inter-dimeric interactions.

To date, a handful of FPC or FPC-associated proteins have been identified (BILBO1, FPC5, FPC4, Tb927.11.5640) [4,9,15,16] but only two, BILBO1 and FPC4, have been characterized at the molecular level [4,10,13,15]. We have previously demonstrated that FPC4 binds to microtubules *via* its N-terminus, and interacts with the NTD of BILBO1 *vi*a its C-terminal domain (CTD). In *T. brucei*, FPC4 localizes at the interface between the FPC and the HC, suggesting a role in linking the FPC-HC-MTQ structures. Static light scattering experiments demonstrated that the BILBO1-NTD and the FPC4-CTD form a stable binary complex [14]. Using a combination of biophysical and cell biology approaches, we have shown that FPC4 binds to the horseshoe-like aromatic patch of the BILBO1-NTD [15].

Despite extensive structural and functional studies on BILBO1, the mechanisms underlying the macro-molecular assembly and biogenesis of the FPC remain elusive, mainly due to the poor knowledge of its molecular composition and assembly. Here we report the identification and characterization of Tb927.6.3240, a novel BILBO1-partner protein at the FPC. We show that it is a *bona fide* FPC protein with an N-terminal domain structurally and functionally homologous to that of BILBO1. We thus named it BILBO2. We demonstrate here that similarly to BILBO1, BILBO2 interacts with the CTD of FPC4 *via* its NTD. We have further determined a 1.9-Å resolution crystal structure of the BILBO2-NTD/FPC4-CTD complex, which provides a clear view of how the extended FPC4 polypeptide docks into the horseshoe-like aromatic pocket of BILBO2. This is the first molecular structure of the FPC protein complex and represents a significant step toward deciphering the FPC interactome in *T. brucei*. Overall, our data identify a common module in two different FPC components that are essential in FPC biogenesis and cell viability.

## Results

### BILBO2 (Tb927.6.3240) is a bona fide FPC protein interacting with BILBO1

Using BILBO1 as a bait in a *T. brucei* 927 genomic yeast two-hybrid screen (Y2H), we identified FPC4 as a BILBO1 binding partner [15]. We also identified a kinetoplastid-specific protein of 271 amino acids, Tb927.6.3240 [17], which we named BILBO2 based on its shared characteristics with BILBO1 (Fig 1A).

The interaction between BILBO1 and BILBO2 identified during the Y2H genomic screen was further narrowed down to sub-domains of the two proteins by serial truncation analyses. Y2H assays demonstrated that aa 151–271 domain of BILBO2 is both necessary and sufficient for the BILBO1-BILBO2 interaction, and was therefore named BILBO1-binding domain (B1BD) (Fig 1B). Interestingly, the Y2H assay also suggests that BILBO2 interacts with other BILBO2 molecules suggesting that BILBO2 could form a homodimer (Fig 1B, BILBO2 x BILBO2).

We previously used the U-2 OS mammalian heterologous expression system as a tool to characterize the BILBO1 polymers and its binding to FPC4 [9,15]. Similarly, we further characterize the BILBO1 domain involved in the interaction with BILBO2. Individual expression of BILBO1 and of HA-tagged BILBO2 (HABILBO2) followed by whole-cell immunodetection shows the BILBO1 polymers (Fig 2A, a) and a cytosolic localization for HABILBO2 (Fig 2A, b).

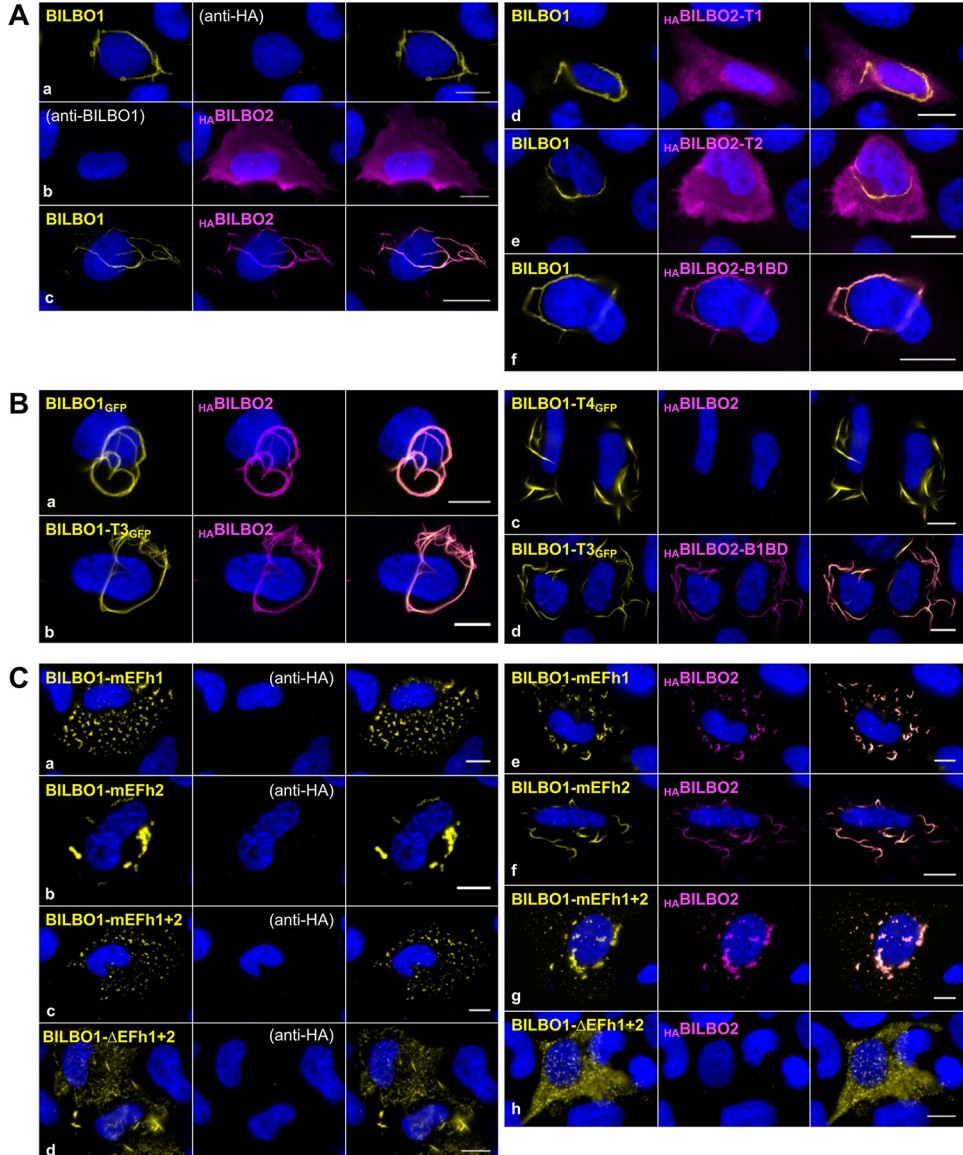

**Fig 2. Analysis of BILBO1 and BILBO2 interaction in a heterologous system. A.** U-2 OS whole cells expressing BILBO1 (a), HABILBO2 (b), or co-expressing BILBO1 with HABILBO2 (c) or with HABILBO2-T1 (d), T2 (e), BILBO2-B1BD (f) domains, and processed for immunofluorescence. **B**. Immunolabeling on detergent-extracted U-2 OS cells co-expressing HABILBO2 and BILBO1GFP (a), BILBO1-T3GFP (b), BILBO1-T4GFP (c), and BILBO1-T3GFP with HABILBO2-B1BD (d). **C.** Immunolabeling on detergent-extracted U-2 OS cells expressing BILBO1 mutated on the EF-hands (mEFh1, mEFh2, mEFH1+2, a-c), deleted of the Efh1+2 domain (d), and co-expressed with HABILBO2 (e-h). Scale bars, 10 μm.

However, co-expression of BILBO1 and HABILBO2 confirmed the interaction between the two proteins as shown by the localization of BILBO2 onto the BILBO1 polymers (Fig 2A, c).

To further characterize the BILBO1-BILBO2 interaction, we individually expressed in U-2 OS cells the domains HABILBO2-T1 (aa 1–103, NTD), T2 (aa 1–150, NTD and linker), and B1BD (aa 151–271) (S1 Fig) individually or in combination with BILBO1 (Fig 2A, d-f). Immunolabelling of BILBO1 and HABILBO2 domains confirms that BILBO2 binds to BILBO1 specifically *via* its BILBO1-BD (B1BD) and not *via* its T1 or T2 domains. BILBO1-BILBO2

interaction is resistant to non-ionic detergent extraction as observed in the extracted cytoskeletons upon co-expression of BILBO1$_{GFP}$ and $_{HA}$BILBO2 (Fig 2B, a). We further used the polymerizing properties of the BILBO1 truncations T3$_{GFP}$ and T4$_{GFP}$ (Fig 1A) [9] to identify the BILBO1 domain involved in the interaction using detergent-extracted cells (Fig 2B). Thereby, we show that BILBO2 interacts with BILBO1-T3 (Fig 2B, b) but not with T4 (Fig 2B, c) suggesting that BILBO2 interacts with the BILBO1 EF-hand domain and that BILBO2-B1BD is sufficient to bind to BILBO1-T3 (Fig 2B, d).

Mutation of key residues for calcium-binding of BILBO1-EFh1 (mEFh1) or EFh2 (mEFh2) induces conformational changes in the shape of the BILBO1 polymers in U-2 OS cells and trypanosomes, suggesting that the *holo* or *apo* calcium status of the EF-hands may regulate BILBO1 assembly [9,10]. We thus wondered if the BILBO1-BILBO2 interaction is modulated by calcium. As previously reported [9], expression of mutated EFh1 (mEFh1) induced the formation of highly compact BILBO1 polymer structures, while mEFh2 and mEFh1+2 induced helical/ball-like structures (Fig 2C, a-c). We further show here that deletion of both EF-hands (ΔEFh1+2) also induce the formation of short polymers (Fig 2C, d). None of the EF-hand mutations affected BILBO2 binding (Fig 2C, e-g). However, their deletion (ΔEFh1+2) abolished the BILBO1-BILBO2 interaction (Fig 2C, h). Interestingly, we observed that, when bound to BILBO2 (Fig 2C, e), BILBO1-mEFh1 formed short linear filaments contrary to the very compact or aggregated structures in the absence of BILBO2 (a). Similarly, mEFh2, when bound to BILBO2 (Fig 2C, f), also formed linear filaments that were not observed when BILBO1-mEFh2 is expressed alone (Fig 2C, b).

In a global quantitative phosphoproteomic analysis, serine 171 and 173 of BILBO2 were shown phosphorylated *in vivo* [18]. Because these residues belong to the BILBO1-BD of BILBO2 (Fig 1A), we postulated that their mutation might affect binding. We co-expressed in U-2 OS cells the non-phosphorylable (S171A, S173A, S171A+S173A) or phosphomimetic (S171D, S173D, S171D+S173D) mutant forms of $_{HA}$BILBO2 with BILBO1. None of these mutations impaired the co-localization of BILBO1 and BILBO2, suggesting that they are not directly involved in the interaction. Nevertheless, the binding affinity might be subtly changed *in vivo*.

Taken together, these data identify BILBO2 as a novel BILBO1-binding protein that plays a role in the shape of the polymers formed by BILBO1 and may regulate BILBO1-mediated FPC assembly or function in the parasite.

## BILBO2 localizes at the FPC primarily via its BILBO1-binding domain

To determine the localization of BILBO2 in *T. brucei*, we generated a guinea pig BILBO2-specific antibody and cell lines expressing epitope-tagged fusion BILBO2 ($_{epitope}$BILBO2) using the pPOTv7 vector series for endogenous tagging [19] and controlled that the tag has no effect on cell growth (S2A Fig). Co-labelling of BILBO1 and BILBO2 on *T. brucei* detergent-extracted cells (cytoskeleton, CSK) revealed that BILBO2 and $_{TY1}$BILBO2 co-localized with BILBO1 at every stage of the cell cycle of PCF (Fig 3A and 3B) and of BSF (S2B Fig). To identify the FPC targeting domain of BILBO2, we generated PCF cell lines inducible for the ectopic expression BILBO2$_{HA}$ and truncations BILBO2-T1$_{HA}$, BILBO2-T2$_{HA}$ and BILBO2-B1BD$_{HA}$. No dominant-negative phenotype or change in cell growth was observed after expression of any of these constructs (S3A Fig). Due to the ectopic expression system, BILBO2$_{HA}$ was expressed at higher level than WT level as observed by western blot analysis, and a large pool was removed during extraction (S3B Fig). This pool was observed in the cytosol by IF and removed during detergent extraction (Fig 3C). A cytoplasmic pool was also observed with the BILBO2-T1$_{HA}$ and BILBO2-T2$_{HA}$ constructs, but to a lesser extent for BILBO2-B1BD$_{HA}$ that

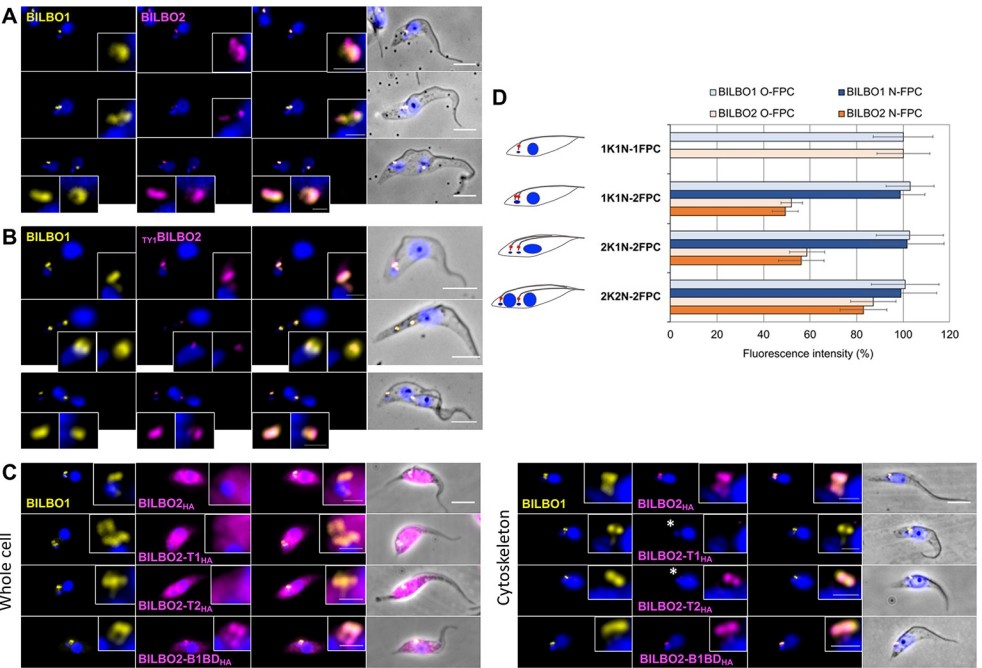

**Fig 3. Cellular localization of BILBO2. A.** Immunolabelling on detergent-extracted PCF cells using anti-BILBO1 and anti-BILBO2 antibodies. **B.** Immunolabelling on detergent-extracted PCF cells expressing endogenously tagged TY1BILBO2 using anti-BILBO1 and anti-TY1 antibodies. **C.** Ectopic expression of BILBO2HA and domains was induced for 24H with 1 μg/mL of tetracycline followed by immunofluorescence on whole cells and detergent-extracted cells (Cytoskeleton). Unlike BILBO2-T1HA, faint but consistent labelling of BILBO2-T2HA the FPC on cytoskeleton is indicated by the asterisk and are to be compared in the insets with increased contrast. Scale bars in A, B and D represent 5μm, and 1 μm (insets). **D.** Sum intensity per collar quantification of BILBO1 and of TY1BILBO2 labelling at the 1K1, 2K1N and 2K2N stages of the PCF cell cycle. Error bars represent the standard error (n = 200).

was less expressed (S3C Fig). Protein localization was assessed by immunofluorescence on WC and CSK, which showed that both BILBO2HA and BILBO2-B1BDHA were detected at the FPC in extracted cells (Fig 3C). The BILBO2-T1HA domain was removed during extraction, suggesting that it is not sufficient for FPC binding. Unlike for BILBO2-T1HA, very weak but consistent BILBO2-T2HA labelling was observed at the FPC (Fig 3C, asterisks in the main panels with increased contrast in the enlarged insets, and to be compared with BILBO2-T1HA), suggesting that the linker domain between the NTD and B1BD may be involved in FPC binding *via* another partner or that the T1 construct was too short for proper function.

Using DAPI as a marker for cell cycle stages (number of kinetoplasts and nuclei) and anti-BILBO1 and anti-TY1 tag, BILBO1 and TY1BILBO2 levels were quantified on CSK PCF cells in four cell cycle stages using ImageJ (Fig 3D). The fluorescence intensity of BILBO1 remained constant at each FPC (old and new) during the cell cycle. Interestingly, the intensity of BILBO2 labelling varied dramatically during the cell cycle, with approximately 50% reduction in cells with 1 kinetoplast, 1 nucleus, but 2 FPCs (1K1N-2FPC, 2K1N-2FPC) in both the new and the old FPC. When the cell reached cytokinesis, the BILBO2 levels were almost equivalent to those at the beginning of the cell cycle. This suggests that BILBO2 expression is cell-cycle regulated in a different way to BILBO1.

To further assess BILBO2 function in PCF and BSF of trypanosomes, we generated BILBO2 RNAi knockdown cell lines in the endogenously tagged background using the tetracycline-inducible RNAi system [20]. PCF and BSF cell growth were not affected after several days of induction, and no morphological phenotypes were observed despite the specific reduction of

BILBO2 expression was observed by western blotting (S4 Fig) suggesting that BILBO2 may not be essential. However, several attempts to generate BILBO2 knock-out PCF and BSF cell lines failed. It is noteworthy that previous RNAi screen indicated that depletion of BILBO2 causes a deleterious effect on cell viability in both trypanosome life forms [21]. These data suggest that BILBO2 might play a critical role in the cell, whereas trace amount of proteins left in RNAi knockdown may have been sufficient to carry out such function and thus showed no defects in cell growth or other phenotypes.

## Localization of BILBO2 to the FPC depends on BILBO1

We have reported previously that, upon RNAi knockdown of BILBO1, PCF cells display a cell cycle arrest in 2K2N stage and a detached new flagellum. Furthermore, the biogenesis of the new FAZ, FP, and FPC are also prevented, which eventually lead to cell death [4]. In addition, some of the FPC4 and MORN1 (a HC protein) proteins are mis-localized within the new detached flagellum [15]. Therefore, to analyse the fate of BILBO2 in the absence of a new FP and a new FPC, we generated an inducible BILBO1 RNAi cell line expressing $_{TY1}$BILBO2 (Fig 4). BILBO1 RNAi-induced cells stopped proliferating after 48h, and displayed a 2K2N growth arrest and detached new flagella (Fig 4A and 4B). Immunofluorescence on whole cells revealed that after 48h of induction BILBO2 was neither detected at the old flagellum nor at the new flagellum. Instead, a punctate cytosolic pool was observed (Fig 4B). Interestingly, western-blotting quantification showed that the CSK-associated pool of BILBO2 decreased by 5.3 x fold over the time-course of BILBO1 RNAi knockdown, while the total pool of BILBO2 (WC) increased by 2.6 x fold (Fig 4C and 4D). These data show that during BILBO1 knockdown, and thus in the absence of a new FPC, BILBO2 is still neo-synthetized but is no longer associated with the cytoskeleton.

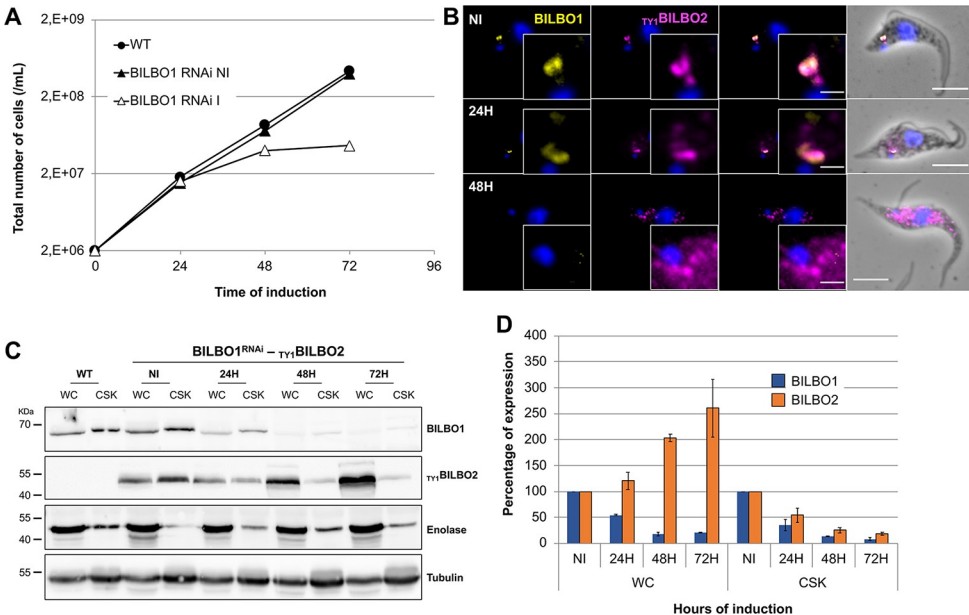

**Fig 4. Depletion of BILBO1 induces cytosolic localization of BILBO2. A.** Comparative growth curves between WT cells and cells expressing $_{TY1}$BILBO2 and non-induced (NI) or induced (I) for *BILBO1* RNAi. **B.** Immunolabelling of BILBO1 and $_{TY1}$BILBO2 on *BILBO1* RNAi non-induced (NI) or induced 24H and 48H whole cells. **C.** Representative western-blot analysis of the fate of BILBO2 during *BILBO1* RNAi in whole-cell (WC) and detergent-extracted samples (CSK). Anti-enolase and anti-tubulin were used as detergent extraction and loading controls, respectively. **D.** Quantification of the Western-blot in C. Error bars represent the standard error from two independent experiments. Scale bars in B represent 5 μm.

## The N-terminal domain of BILBO2 is homologous to the BILBO1 N-terminal domain

BILBO1 is a multi-domain protein and its NTD interacts with FPC4-CTD *via* key residues in a conserved surface patch that are involved in BILBO1 function [15]. Database mining for proteins sharing a domain homologous to BILBO1-NTD identified the N-terminal domain of BILBO2. Alignment of BILBO1 and BILBO2 sequences revealed an overall similarity of 19% between the full-length proteins. However, the identity and similarity reach 32% and 38% respectively between their NTD (Fig 5A). Importantly, the two residues (Y64, W71), which were previously shown to play a critical role in BILBO1's interaction with FPC4 and cell viability [9,13–15], are conserved or identical in BILBO2 (*i.e.* F63, W70). However, apart from its NTD and B1BD, no other structural or functional domains were identified in BILBO2.

We hypothesized that if the NTD of BILBO1 and BILBO2 are functionally similar, they could be exchanged without affecting the function of either protein *in vivo*. Recoded chimeric BILBO1 and BILBO2 proteins with exchanged NTDs, namely $_{Ch}$BILBO1-BILBO2$_{HA}$ (BILBO1 aa1-118 fused to BILBO2 aa111-271) and $_{Ch}$BILBO2-BILBO1$_{HA}$ (BILBO2 aa1-110 fused to

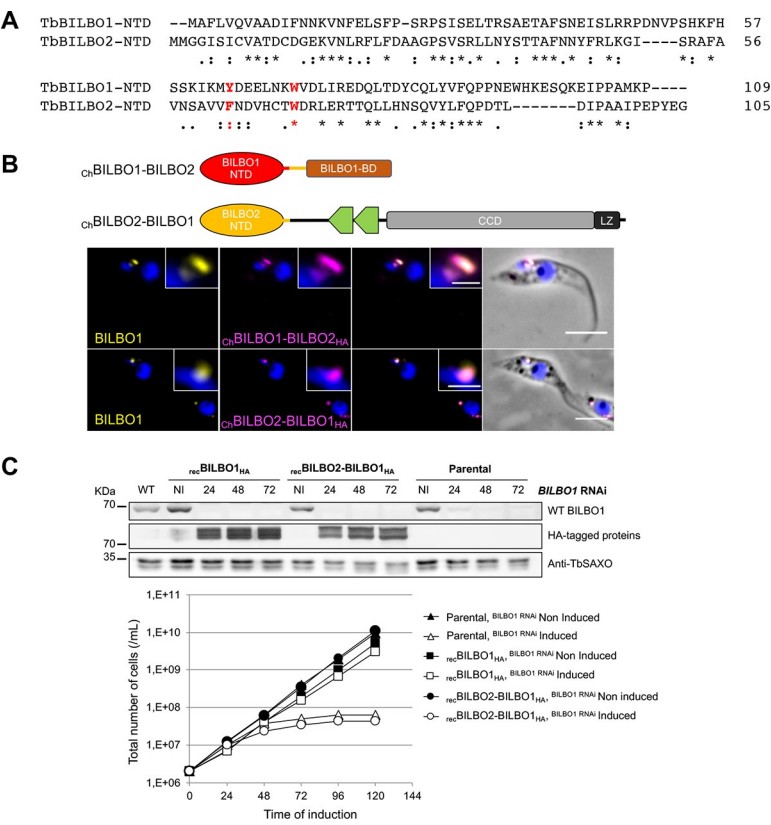

**Fig 5. BILBO1 and BILBO2 share a conserved N-terminal domain with conserved residues. A.** Alignment of the BILBO1 and BILBO2 NTD domains. Asterisks indicate identical residues; colons indicate conserved substitution; periods indicate semi-conserved substitutions. **B.** Immunolocalization of chimeric BILBO1-BILBO2 proteins. Anti-BILBO1 labels both BILBO1 and $_{Ch}$BILBO1-BILBO2$_{HA}$; anti-HA labels $_{Ch}$BILBO1-BILBO2$_{HA}$ or $_{Ch}$BILBO2-BILBO1$_{HA}$. Cells were induced 18h with 1 µg.mL$^{-1}$ tetracycline and detergent-extracted for immuno-labelling. Scale bars represent 5 µm. **C.** Cells inducible for *BILBO1* RNAi and for the expression of recoded $_{rec}$BILBO1$_{HA}$ or $_{rec}$BILBO2-BILBO1$_{HA}$ were induced with 2 µg.mL$^{-1}$ of tetracycline. The parental cells are inducible for *BILBO1* RNAi only. Top panel: Western blot analysis of whole cells non induced or induced at different time points. Anti-TbSAXO was used as loading control. Bottom panel: Growth curves of non-induced and induced cells showing that $_{rec}$BILBO1$_{HA}$ can rescue the RNAi growth defect, contrary to $_{rec}$BILBO2-BILBO1$_{HA}$.

BILBO1 aa119-587), were ectopically expressed in PCF *T. brucei* (Fig 5B). It is important to note that, as previously described [9], ectopic long-term and/or high level expression of WT BILBO1, T3 or T4 was lethal due to excessive polymer formation induced by the CCD and LZ domains [9]. This explains the growth phenotype that occurs when expression of BILBO1$_{HA}$ and of $_{Ch}$BILBO2-BILBO1$_{HA}$ is induced (S5 Fig). Nevertheless, using an anti-BILBO1 antibody recognizing aa1-110 of both BILBO1 and the chimeric $_{Ch}$BILBO1-BILBO2$_{HA}$ proteins, and an anti-HA antibody recognizing the chimeric proteins only, we immuno-localized endogenous BILBO1 and both chimeric proteins in detergent-extracted cells. Both $_{Ch}$BILBO2-BILBO1$_{HA}$ and $_{Ch}$BILBO1-BILBO2$_{HA}$ targeted to the FPC and, apart from the CCD-related problem mentioned above, no specific cell growth or morphology phenotype was observed (Figs 5B and S5). This supports the hypothesis that the NTDs are not involved in cellular localization but that they could have a similar functional role. The recoded BILBO1$_{HA}$ and $_{Ch}$BILBO2-BIL-BO1$_{HA}$ constructs (*i.e.* resistant to the *BILBO1* RNA interference) were also expressed in the BILBO1 RNAi background and were still expressed after *BILBO1* RNAi induction (Fig 5C, western blot panel). However, whilst expression of recoded BILBO1$_{HA}$ could rescue the *BILBO1* RNAi growth phenotype (Fig 5C, growth curve panel), recoded $_{Ch}$BILBO2-BILBO1$_{HA}$ could not, and cells stopped growing in a similar time frame as the cells induced for *BILBO1* RNAi only (Parental cells). This suggests that the NTDs are similar but are not identical domains that are not interchangeable *in vivo*.

## BILBO2 is an FPC protein as well as a FPC4 interacting partner and binds to FPC4 via its N-terminal domain

We previously demonstrated that FPC4 binds to microtubules *via* its N-terminus, and interacts with BILBO1 *via* its C-terminal region (FPC4-B1BD), suggesting a role in linking the FPC-HC-MTQ structure. Using a combination of site-directed mutagenesis, biophysical and cell biology approaches, we also showed that FPC4 binds to a conserved hydrophobic patch on the BILBO1-NTD surface patch. Further, static light scattering experiments demonstrated that the BILBO1-NTD and the FPC4-B1BD form a stable binary complex [15]. The recent high-resolution crystal structure of BILBO1-NTD and mutagenesis studies revealed that FPC4 interacts with BILBO1 by mainly contacting three aromatic residues W71, Y87, and F89 in the centre of the conserved hydrophobic patch [14].

Based on the high homology between NTDs of BILBO1 and BILBO2, we checked whether BILBO2-NTD also forms a stable complex with FPC4-B1BD. We first tested the interaction between BILBO2 and FPC4 (full-length or truncations) by Y2H (Fig 6A). Their interaction was confirmed with both full-length sequences. Further, BILBO2-NTD and FPC4-B1BD domains are both necessary and sufficient for the interaction. This was also supported by the proximity of FPC4 and BILBO2 in *T. brucei* immuno-fluorescence labelling at different stages of the cell cycle (Fig 6B), as it was observed for BILBO1 and FPC4 [15]. Because triple labelling of BILBO1, BILBO2 and FPC4 was challenging, probably due to primary and secondary antibodies steric hindrance, we turned to ultrastructure expansion microscopy (U-ExM) that allows the expansion of a sample and the visualization of preserved ultrastructure of macromolecules by optical microscopy [22,23]. This approach facilitated the localization of the FPC (labelled with BILBO1) in respect to the MTQ and to the axoneme that goes through (S6A Fig). The MTQ extends from between the mature and the immature basal bodies, turns around the FP and is prolonged along the FAZ as previously described in [5,6]. Higher resolution of the BILBO1 and BILBO2 colocalization was further determined using the same approach coupled to confocal microscopy (S6B Fig and S1 Movie). These approaches allowed us to evidence the annular shape of the FPC and the overall co-localization of BILBO1 and BILBO2 at the

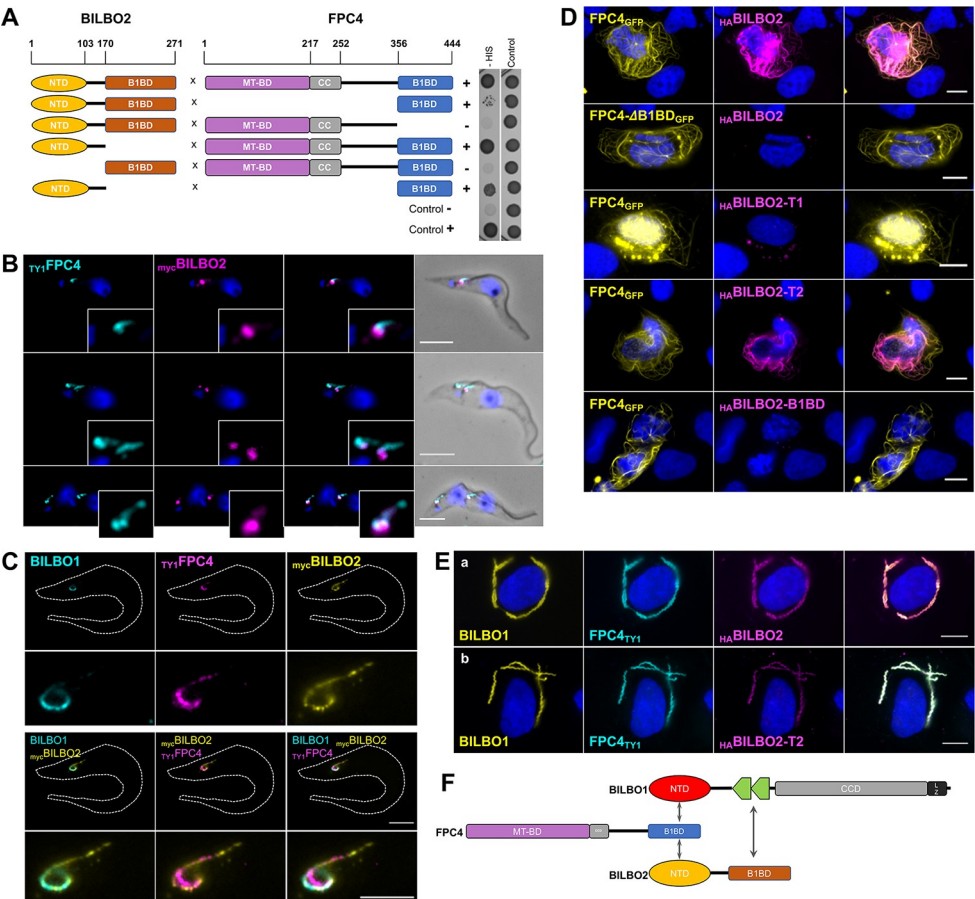

**Fig 6. Interaction between BILBO2 and FPC4 is similar to that of BILBO1 and FPC4. A.** Y2H interaction assay between BILBO2 and FPC4 and domains. **B.** Immuno-colocalisation of TY1FPC4 and BILBO2 in detergent-extracted PCF cells. Scale bars 5 μm. **C.** Immunolocalization of BILBO1, mycBILBO2 and TY1FPC4 using U-ExM in detergent-extracted PCF cells. Scale bars 10 μm and 5 μm in enlarged inset. **D.** Expression in U-2 OS cell and immunolocalization of FPC4GFP and FPC4 deleted of its B1BD (FPC4-ΔB1BDGFP) and HABILBO2 and domains demonstrating that the BILBO2-T1 (aa 1–103) domain is not sufficient for a stable interaction between BILBO2 and FPC4 whereas a longer domain (BILBO2-T2) is stabilizing the interaction. Cells were detergent-extracted before the IF to reduce the FPC4 and BILBO2 cytosolic labelling. Scale bars 10 μm. **E.** A tripartite interaction is demonstrated in U-2 OS cells by the co-labelling of FPC4 and BILBO2 and domains onto the BILBO1 polymers. Scale bars 10 μm. **F.** Schematic representation of the interactions between BILBO1, BILBO2 and FPC4.

FPC. Further, triple labelling of BILBO1, BILBO2 and FPC4 showed that FPC4 partially colocalizes at the FPC and extends on the shank of the Hook Complex with a regular pattern as previously demonstrated in [15] (Fig 6C). Interestingly, BILBO2 colocalizes with BILBO1 at the FPC following also a regular pattern that extends past the FPC where it colocalizes with FPC4.

We then took advantage of the property of FPC4 to bind to MT [15] to assess the localization of HABILBO2 in detergent-extracted U-2 OS cells expressing FPC4GFP (Fig 6D). When co-expressed, BILBO2 localized onto the MT labelling of FPC4, confirming their specific interaction *in vivo*. The deletion of the FPC4-B1BD abolished the interaction and resulted in the removal of BILBO2 during detergent extraction. Interestingly, BILBO2-T2 (aa1-151) can bind to FPC4, whereas BILBO2-T1 (aa 1–103) was extracted, suggesting that this construct might be too short for correct folding or disrupts the binding site for FPC4. Finally, the BILBO2-B1BD does not bind to FPC4 and is removed during extraction.

Because BILBO2 binds to BILBO1 (*via* its CTD) and to FPC4 (*via* its NTD), we tested whether a tripartite interaction could occur *in vivo* (Fig 6E). Immuno-labelling of co-expressed BILBO1, HABILBO2, and FPC4TY1 in U-2 OS cells demonstrated that both FPC4 and BILBO2 can bind to the BILBO1 polymers (Fig 6E, a). This triple co-localization was also observed when BILBO2 deleted for its BILBO1 binding domain (BILBO2-T2, which binds to FPC4 but not to BILBO1) was expressed (Fig 6E, b) suggesting a tripartite interaction schematized in Fig 6F.

## Crystal structure of the BILBO2-NTD/FPC4-CTD complex provides the first atomic view of two FPC components

To reveal the molecular mechanisms underlying BILBO2-FPC4 interaction, we have determined a crystal structure of the BILBO2-NTD/FPC4-B1BD complex (Fig 7). The crystal structure was determined at 1.86 Å resolution by molecular replacement using the structure of BILBO1-NTD (6JSQ.pdb) as the search model (Table 1). There are two copies of the heterodimer *per* asymmetric unit in the crystal. Both structures contain aa4-110 of BILBO2 and aa432-

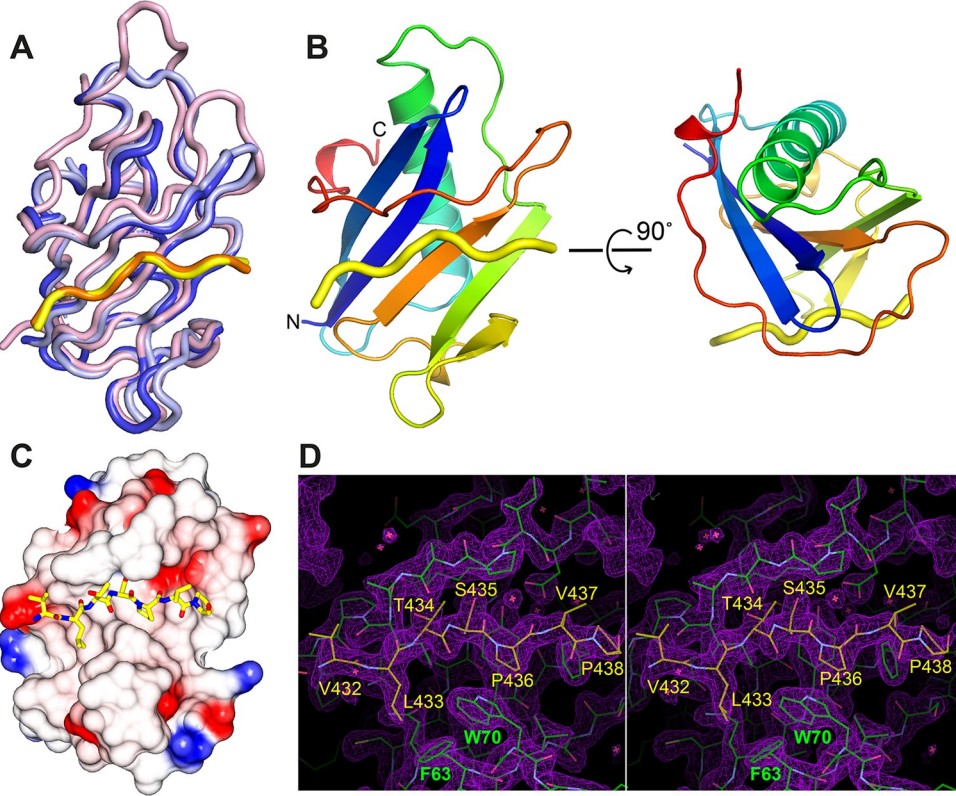

**Fig 7. Crystal structure the BILBO2-NTD/FPC4-B1BD complex. A.** Superposition of the two copies of BILBO2-NTD in the asymmetric unit cell of the crystal lattice (blue and light blue), together with BILBO1-NTD (pink, 6SJQ.pdb). Shown in yellow and orange are polypeptides of FPC4 bound to BILBO2-NTD. **B.** Ribbon diagram of the structure of the BILBO2-NTD/FPC4 complex in two orthogonal views. The structure of BILBO2-NTD is colour-ramped from blue to red at the N- and C-termini, respectively. FPC4 is shown as a yellow tube. **C.** Crystal structure of the complex with FPC4 residues depicted as sticks and BILBO2-NTD as an electrostatic surface plot. **D.** Stereo view of the $2F_o$-$F_c$ map (purple) around the binding interface between FPC4 and BILBO2 contoured at 1.5 σ level. BILBO2 and FPC4 are coloured in green and yellow, respectively. All visible residues of FPC4 (aa 432–438) and two interface aromatic residues of BILBO2 (F63, W70) are labelled. Plots in (A) and (B) were generated using PyMOL (*The PyMOL Molecular Graphics System*, *Version 1.2r3pre*, *Schrödinger, LLC.*), the one in (C) was done by CCP4mg Presenting your structures: the CCP4mg molecular-graphics software [43], and that in (D) by COOT [39].

**Table 1. Data collection and refinement statistics.**

| Data collection | |
|---|---|
| Space group | C222$_1$ |
| Wavelength (Å) | 0.976 |
| Cell dimensions | |
| *a*, *b*, *c* (Å) | 60.42, 73.39, 120.51 |
| Resolution (Å) | 50.0–1.87 (1.94–1.87) * |
| Total reflections | 271,309 (15,723) |
| Unique reflections | 21,911 (1,295) |
| Multiplicity | 12.4 (9.7) |
| R-merge | 0.125 (2.359) |
| R-meas | 0.130 (2.477) |
| R-pim | 0.037 (0.727) |
| CC(1/2) | 0.999 (0.742) |
| CC* | 1.000 (0.923) |
| Mean I / $\sigma(I)$ | 12.76 (0.77) |
| Completeness (%) | 95.4 (57.7) |
| Wilson B-factor | 39.35 |
| **Refinement** | |
| Resolution (Å) | 20–1.87 |
| Number of reflections | 21,481 |
| $R_{\text{work}}$ / $R_{\text{free}}$ (%) | 21.4/25.1 |
| No. atoms | |
| Protein | 1,765 |
| Water | 82 |
| Ligand/ion | 36 |
| *B*-factors | |
| Protein | 52.0 |
| Water | 45.2 |
| Ligand/ion | 77.8 |
| R.m.s. deviations | |
| Bond lengths (Å) | 0.007 |
| Bond angles (°) | 0.810 |
| Ramachandran plot | |
| Favored (%) | 99.07 |
| Allowed (%) | 0.93 |
| Outlier (%) | 0 |

*Values in parentheses are for the highest resolution shell.

438 of FPC4, except aa26-27 that were missing in one BILBO2 molecule. The two structures of the complex are nearly identical, with a root-mean-square deviation (RMSD) of 0.46 Å for all aligned backbone atoms (Fig 7A). These two structures of BILBO2-NTD are also very similar to our previously reported crystal structure of BILBO1-NTD [14], with RMSD values of 0.84 Å and 0.98 Å. Similar to the BILBO1-NTD structure, the C-terminal part of BILBO2-NTD (aa91-110) also forms a horseshoe-like loop that wraps around the first two β strands and the long α helix of the core structure (Fig 7B). FPC4 is docked into the hydrophobic cleft between the C-terminal loop and the β sheet (Fig 7C). Interestingly, despite the fact that we used aa394-444 of FPC4 to prepare the sample for crystallization, only residues L432-P438 were visible in

the final structure, suggesting that the rest of FPC4 does not bind to BILBO2 and is thus disordered as predicted (Fig 7D). This part of FPC4, particularly residues S435 and P436 in the centre of the stretch, makes extensive contacts with BILBO2 through multiple hydrogen bonds and many hydrophobic interactions (S7 Fig). This structure represents the first-ever high-resolution view of intermolecular interactions between two FPC components.

## Mutation of critical interface residues abolishes the BILBO2-FPC4 interaction

A close look in the complex structure suggests that residues S435, P436, and P438 of FPC4 are in close contact with BILBO2 (Fig 8A). We thus mutated each of these three residues to alanine and checked whether the mutants affect the interaction. Firstly, isothermal titration calorimetry (ITC) experiments with BILBO2-NTD and FPC4-B1BD demonstrated interaction with a dissociation constant (Kd) of 2.42 μM (Fig 8B), which suggested a slightly tighter interaction than that between BILBO1-NTD and FPC4 (Kd = 5.4 μM) [15]. For the three mutants, two of them (*i.e.* S435A and P436A) completely abolished the BILBO2-NTD/FPC4-B1BD interaction, whilst mutant P438A only marginally affected the interaction (Kd = 4.31 μM). The role of S435 and P436 in the interaction was additionally confirmed in the U-2 OS system. BILBO2, which normally binds to FPC4 in these cells (Fig 6C), was detergent-extracted from FPC4-S435A_GFP and FPC4-P436A_GFP expressing cells demonstrating the lack of interaction (Fig 8C, a).

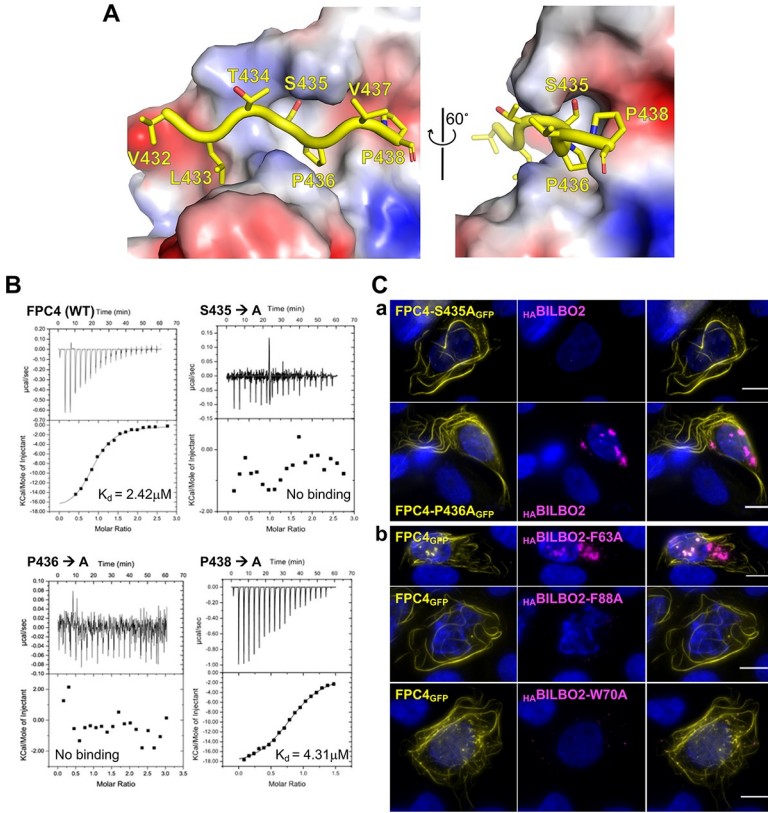

**Fig 8. Identification of key residues involved in the BILBO2-FPC4 interaction. A.** Interaction between FPC4-B1BD and BILBO2-NTD. FPC4 residues are depicted as sticks in yellow, whereas BILBO2-NTD is shown as an electrostatic surface plot. **B.** ITC experiments using purified BILBO2-NTD and wild-type (WT) and mutants of FPC4-B1BD. **C.** Co-immunolocalization of _HA_BILBO2 (magenta) and of mutated forms of FPC4_GFP (yellow) in U-2 OS cells (a). Co-immunolocalization of FPC4_GFP (yellow) and mutated forms of _HA_BILBO2 (magenta) (b). Scale bars, 10 μm.

We have previously identified several key aromatic residues in BILBO1-NTD that are important for the BILBO1 function and involved in the FPC4-BILBO1 interaction (residues Y64, W71, Y87, F89) [14,15]. The crystal structure of the BILBO2-FPC4 protein complex also suggests that residues F63, F88, and W70 in the BILBO2-NTD are involved in the interaction with FPC4. Here we show in U-2 OS cells that substituting residues F63, F88, or W70 in BILBO2 to alanine indeed abolished the interaction *in vivo* (Fig 8C,b).

To gain further insights into BILBO1-BILBO2-FPC4 tripartite dependencies and hierarchies, we generated cell lines expressing endogenously-tagged $_{TY1}$FPC4 and $_{myc}$BILBO2 in either the inducible *BILBO2* or *FPC4* RNAi knockdown background. We then analysed the fate of the proteins by IF and WB. As previously shown [15], FPC4 RNAi did not induce any cell growth phenotype (S8A Fig). Further, when *FPC4* RNAi is induced neither BILBO2 expression levels nor localization were affected (S8B and S8C Fig). Similar results were also observed for FPC4 in the *BILBO2* RNAi background whose expression level remained constant and localized to the cystoskeleton (S8 Fig) suggesting that expression and localization of FPC4 and BILBO2 are not interdependent.

## Discussion

The FPC is a kinetoplastid-specific cytoskeleton structure at the neck of the FP. Although the FPC is essential and required for the biogenesis of the FP and cell survival, its molecular composition and function are unknown. We previously provided the first molecular evidence that FP biogenesis is mediated by the cytoskeleton *via* the FPC protein BILBO1 (4). Based on previous work on the polymerization properties of BILBO1 and its interaction with FPC4, a microtubule-binding protein, we hypothesized that BILBO1 forms, at the FPC, a molecular framework upon which other proteins can dock, and that the BILBO1-NTD and EF-hands domains could be anchoring or modulator sites [9–12,15]. Here, we identify BILBO2 (Tb927.6.3240) as a novel FPC component and as a BILBO1 partner protein. We show that BILBO2 interacts with the BILBO1 EF-hand domains *via* its C-terminal domain and with FPC4 *via* its N-terminal domain. Together with BILBO1, BILBO2 is the second *bona fide* FPC protein and is involved in a tripartite interaction with BILBO1 and FPC4 (Fig 6F).

RNAi knockdown of BILBO2 was not lethal suggesting that the protein is not essential. However, it is likely that knockdown was incomplete, a phenomenon that has been described before [24]. Indeed, Fig 3D shows that, compared to 1K1N-1FPC cells, a BILBO2 signal of 50% per flagellar pocket collar in 1K1N-2FPC is sufficient for the FPC to build. These observations support the idea that an incomplete knockdown would not be enough to efficiently disrupt flagellar pocket collar formation or induce other phenotypical effect. Further, repeated failure in producing BILBO2 double knockout in PCF and BSF cell lines supports the essentiality hypothesis and this will be further investigated.

### BILBO2 and BILBO1 interaction

Y2H assays previously showed that FPC5, a putative kinesin, binds to the BILBO1 EF-hands [9]. Here, we showed that BILBO2 also binds to the BILBO1 EF-hands (Fig 1), but the interaction is not affected by mutating the calcium-binding sites (Fig 2). Moreover, the binding of BILBO2 modulates the shape of the BILBO1 polymer (demonstrated by the change from compact to less dense filaments formed by BILBO1 with mutated EF-hands) suggesting that BILBO2 may participate to the plasticity of the BILBO1 polymers. This could be a role for BILBO2 during the cell-cycle. Indeed, BILBO2 FPC-associated protein levels drop by 50% in 1K1N-2FPC cells compared to the 1K1N-1FPC levels while BILBO1 level in each FPC remains constant (Fig 3). Cell cycle regulation of BILBO2 was previously demonstrated by the study of

the PCF cell cycle proteome where BILBO2 level is maximum in late G1 cells and decreases to 30% in G2/M cells, and BILBO1 level remains constant [25]. Variation of the BILBO2 levels at the FPC could thus influence the shape of the BILBO1 polymer and participate in the function or in the biogenesis of the FPC during the cell cycle. However, we have not elucidated the molecular mechanism(s) by which BILBO2 is able to influence BILBO1 filament shape. Further studies of the BILBO1 polymer structure in the absence and in presence of BILBO2 will provide important clues to understand the role of the FPC architecture.

STED microscopy previously showed a dotted pattern parallel to the FPC and along the MTQ on the distal side of the FPC for FPC4 [15]. Interestingly, BILBO2 is also observed as dotted pattern along the MTQ but also at the FPC. Three-dimensional rendering of the expansion microscopy labelling (S6B Fig) indicated that BILBO1 and BILBO2 do not localise to the same microtubules in the distal part of the MTQ. The reason for these different localizations is unknown and will be further studied. Fine localization of BILBO1 and BILBO2 at the FPC and the MTQ is currently under investigation to further understand their relationship.

## Crystal structure of the BILBO2-FPC4 complex

In Y2H assays and U-2 OS cells, the BILBO2 T2 construct (aa1-151) interacts with FPC4 whilst T1 (aa1-103) did not, suggesting that aa104-151 are involved in the interaction. The crystal structure of the BILBO2-NTD (aa1-110) in complex with the FPC4-B1BD (aa354-404) reveals that the aa91-110 form an extended loop wrapping around the first two β strands and the long α helix of the core structure, which helps the protein to form a horseshoe-like hydrophobic pocket for FPC4 binding. This is reminiscent of what was reported for BILBO1-NTD [14]. Thus, the T1 construct (aa1-103) is too short to form the intact binding site for FPC4.

## Functional role of the NTDs

BILBO1 and BILBO2 NTDs share a highly similar three-dimensional structure with conserved residues that are essential for the interaction with FPC4. Interestingly, only five residues (aa432-438) of the FPC4-B1BD (aa354-444) are visible in the crystal structure of the BILBO2-FPC4 protein complex. The stretch of FPC4 binds to the hydrophobic pocket and passes through the gap of the horseshoe gauged by the highly conserved aromatic residue W70. We could confirm the binding by the abolished interaction upon mutating W70, as well as the conserved aromatic residue F88 in the centre of the pocket, to alanine. Conversely, mutation to alanine of the two central residues S435/P436 of the bound peptide completely disrupts the interaction of FPC4 with BILBO2. This is similar to the reported interaction between FPC4 and BILBO1 [15] and suggests that FPC4 might use a similar strategy to bind BILBO1 and BILBO2. However, the binding affinity between FPC4 and BILBO2 (Kd = 2.4 μM) is higher than that between FPC4 and BILBO1 (Kd = ~6 μM) [15]. Such a subtle difference in binding affinities suggests possible hierarchical orders in recognizing the two partner proteins in the parasite. Also, the BILBO1 and BILBO2 NTDs are not interchangeable in the parasite, suggesting that in the parasite's cellular environment the NTDs play different roles.

## The BILBO1-BILBO2-FPC4 tripartite interaction

We have demonstrated that BILBO2 in *T. brucei* is localized at the FPC, is a novel BILBO1 binding protein, and that it is involved in a tripartite interaction with BILBO1 and FPC4 (schematized in Fig 6F). The tripartite interacting domains have been identified, and mutational analyses in U-2 OS cells suggest that BILBO2 can influence the shape of the BILBO1 filaments by interacting with the BILBO1 EF-hand domains.

Further, we demonstrate that BILBO1 and BILBO2 share a structurally similar NTD domain with similar key residues interacting with FPC4. BILBO2-NTD and FPC4 interact in Y2H experiments, in U-2 OS cells, and *in vitro*, but the interaction is very weak in *T. brucei* detergent-extracted cytoskeleton (Fig 3C). The same is true for the BILBO1-NTD constructs and FPC4-B1BD constructs that were not observed at the FPC after detergent-extraction in trypanosomes [9,15]. This suggests either low-affinity interaction *in vivo*, or that the tripartite BILBO1-BILBO2-FPC4 interaction is necessary to dock FPC4 and BILBO2 at the FPC. However, FPC4 localizes at the edge of the FPC and HC structures [15] whilst BILBO2 co-localizes with BILBO1 on the whole FPC structure. This might suggest that BILBO2 interacts with FPC4 on the edge of the FPC and along the MTQ.

The BILBO1-BILBO2-FPC4 tripartite interaction might occur exclusively at the proximal portion of the FPC indicating that other, yet to be discovered, proteins are likely to be involved in this protein complex. It is worth mentioning that the binding affinity of FPC4 to BILBO2 is higher than with BILBO1 [15]. We, therefore, speculate that a hierarchy exists where BILBO1 may allow FPC4 to come into close proximity to BILBO2 and then FPC4 interacts with BILBO2. Notably, BILBO2 also interacts with other BILBO2 molecules in Y2H assays suggesting some degree of dimerization or polymerization can occur. Further, the variation of BILBO2 levels at the FPC during the cell cycle could regulate the interaction between FPC4 and BILBO1 and as a consequence the structure or the function of the FPC.

## Conclusion

BILBO1, BILBO2, and FPC4 clearly interact to potentially form a tripartite complex *in vivo*, and the NTDs of the former two play a significant role in their interaction. It would be important to further understand the role of these NTDs by searching for potential new partners. Overall, our structural and functional data have shed light on the first structure of an FPC protein complex and represent an additional step in deciphering the FPC structure and function in *T. brucei*.

## Material and methods

### Cell lines, culture and transfection

*Trypanosoma*. The trypanosome cell lines used in this study were derived from the parental (WT) procyclic form SmOxPCF427 (PCF) and bloodstream form SmOxBSF427 (BSF), both co-expressing the T7 RNA polymerase and tetracycline repressor [26]. PCF cells were cultured at 27˚C in SDM79 medium (PAA, Ref: G3344,3005) containing 10% (v/v) heat-inactivated foetal calf serum, 10 μg.ml$^{-1}$ Hemin, 26 μM sodium bicarbonate, 10 mM D-Glucose, 3.5 mM L-glutamine, 5.3 mM L-proline, 0.9 mM sodium pyruvate, 3.4 mM L-threonine, 150 μM glutamic acid, 120 μM sodium acetate, 230 μM D-glucosamine, puromycin 1 μg.ml$^{-1}$. BSF cells were cultured at 37˚C and 5% $CO_2$ as described in [19] in IMDM medium containing 10% (v/v) heat-inactivated foetal calf serum, 36 mM sodium bicarbonate, 136 μg.mL-1 hypoxanthine, 39 μg.mL$^{-1}$ thymidine, 110 μg.mL$^{-1}$ sodium pyruvate, 28 μg.mL-1 bathocuproine, 0.25 mM β-mercaptoethanol, 2mM L-cysteine, 62.5 μg.mL$^{-1}$ kanamycin, 0.1 μg.mL$^{-1}$ puromycin. Cells were transfected as described in [27] using an AMAXA electroporator and program X-001 with the transfection buffer described in [28]. After transfection, the cells were selected using the appropriate antibiotic (for PCF and BSF respectively blasticidin 20 μg.mL$^{-1}$ or 10 μg.mL$^{-1}$, phleomycin 5 μg.mL$^{-1}$ or 2.5 μg.mL$^{-1}$, neomycin 10 μg.mL$^{-1}$ or 2.5 μg.mL$^{-1}$). Clones were selected after serial dilutions. Ectopic expression and RNAi were induced with tetracycline at 1 μg.mL$^{-1}$ and 10 μg.mL$^{-1}$, respectively except for Fig 5C were cells were induced with 2 μg.mL$^{-1}$ tetracycline.

*U-2 OS cells* (human bone osteosarcoma epithelial cells, ATCC Number: HTB-96 [29] were grown in D-MEM Glutamax (Gibco) supplemented with 10% foetal calf serum and 1% penicillin-streptomycin at 37˚C plus 5% $CO_2$. Exponentially growing cells were transfected with 0.5–1 μg DNA using Lipofectamine 2000 in OPTIMEM (Invitrogen) according to the manufacturer's instructions and processed for IF 24 h post-transfection.

## Molecular biology, cloning

BILBO2 ORF was initially cloned by PCR using Tb927 genomic DNA as a template.

*Trypanosome vectors.* For the RNAi experiments in PCF and BSF, the *BILBO2* fragment (bp 205–816) was cloned between the *XbaI/XhoI* restriction sites of the p2T7-tiB/GFP vector [30]. The full-length *BILBO2* ORF and truncations were cloned into the *HinIII/XhoI* sites of pFS-1 vector containing a C-terminal 3HA-tag to express BILBO2-HA, BILBO2-T1-HA, BILBO2-T2-HA and BILBO2-B1BD-HA (vector modified from [31]). To generate the chimeric *BILBO1* and *BILBO2* constructs, *BILBO1* RNAi-resistant recoded *BILBO1* and *BILBO2* coding sequences were obtained from Eurofins and used as template in overlapping PCR. The PCR products were then cloned into the pFS-1-3HA vector. For N-terminal 10xTY1 and 10xc-myc tagging of BILBO2, long primers were designed as described in [19] and the PCR was performed with the pPOTv7 vector template. The PCR product was transfected as described in [19].

*Yeast-two hybrid vectors.* Full-length BILBO2 was cloned into the pGBKT7 and pGADT7 vectors and truncations were cloned into the pGBKT7 vector. The *TbBILBO1* and *TbFPC4* vectors are described in [9,15].

## Mammalian vectors

The *BILBO1* ORF was cloned into the pcDNA3 vector as described in [9]. The *BILBO2* ORF or truncations were cloned into the pcDNA3.1 vector, in frame with N-terminal 3xHA tag, from which the mutants for serine 171 and 173 were generated using the QuickChange II site-directed mutagenesis kit (Agilent). The BILBO1, BILBO1GFP, FPC4GFP, FPC4-B1BDGFP and FPC4-ΔB1BDGFP constructs are described in [9,15]. The *FPC4* ORF was also cloned into the pcDNA3.1 vector, in frame with a C-terminal 3xTY1 tag.

## Bacterial vectors

To allow the coexpression of BILBO2-NTD and FPC4-B1BD, two vectors containing different antibiotic resistances were used. Both vectors contain a similar His6-SUMO tag, which was included to increase protein expression and solubility of recombinant proteins [32,33]. The constructs also contain a cleavage site between the tag and the target proteins, which is cut by the highly specific SENP2 (Sentrin/SUMO-specific protease 2) protease. The SUMOpET15b vector, containing ampicillin resistance and NdeI/BamHI cleavage sites, was used to express BILBO2-NTD. The SUMOpET28a vector, containing kanamycin resistance and BamHI/XhoI cleavage sites, was used to express wild-type and mutants of FPC4-B1BD. All mutants were generated by site-directed mutagenesis using a QuikChange kit (Stratagene) according to the manufacturer's instructions. Incorporations of mutations were confirmed by DNA sequencing.

## Production of guinea pig anti-BILBO2 serum

*BILBO2* ORF was cloned into pET28a(+) (NOVAGEN) and expressed in BL21(DE3) *E. coli*, and purified in 8M urea, 50mM NaPi buffer pH7.4, 500mM NaCl. Guinea pig immunization was done by Eurogentec. For western blotting, the antibody was further purified using the

Melon Gel purification kit (Thermo scientific #45206) according to the manufacturer's instructions.

## Immunofluorescence

**Wide-field fluorescence microscopy on *trypanosome*.** Cells were processed and fixed as described in [9], except for BSF that were washed in vPBS (NaCl 0.8 mg.mL⁻¹, KCl 0.22 mg. mL⁻¹, $Na_2HPO_4$ 22.7 g.mL⁻¹, $KH_2PO_4$ 4.4 mg.mL⁻¹, sucrose 15.7 mg.mL⁻¹, glucose 1.8 mg.mL⁻¹), resuspended in 0.25% Nonidet P-40 (IGEPAL), 100 mM PIPES-NaOH pH6.9, 1 mM $MgCl_2$, and loaded on poly-L-lysine-coated slides for 10 min. After either 3% paraformaldehyde (PFA) fixation for 5 min at RT (then 100 mM glycine neutralization) or -20°C methanol fixation for 30 min, the slides were washed in PBS twice for 5 min and incubated with primary antibodies in PBS for 1 h in a dark moist chamber (anti-BILBO1 1–110 [9], 1:4,000 dilution; anti-HA tag IgG1 mouse monoclonal, Biolegend 901514, 1:1000; anti-HA IgG2a mouse monoclonal, GeneTex GTX628902, 1:2000; mouse anti-TY1 tag BB2 IgG1, 1:1000 [34]; anti-BILBO2, 1:100). Following the primary antibody incubation, samples were washed twice (5 min) in PBS and incubated 1h with the secondary antibodies [anti-mouse IgG conjugated to FITC (Sigma F-2012, 1:100); anti-rabbit IgG conjugated to Alexa fluor 594 (Molecular Probes A-11012, 1:100); anti-mouse IgG1 conjugated to Alexa fluor 594 (Molecular Probes A-21125, 1:100); anti-mouse IgG2a conjugated to Alexa fluor 488 (Molecular Probes A-21131, 1:100); anti-rabbit conjugated to Alexa fluor 647 (Molecular Probes A-31573, 1:100); Alex fluor488--conjugated anti-guinea pig (Molecular Probes A-11073, 1:100)] according to the primary antibody combination. After two 5 min washes in PBS, kinetoplasts and nuclei were labelled for 5 min with DAPI (10 μg.mL⁻¹) followed by two PBS washes. Slides were mounted with SlowFade Gold Kit (Molecular Probes, S-36936).

**Wide-field fluorescence microscopy on U-2 OS cells.** U-2 OS cells grown on glass coverslips were washed with PBS, fixed in 3% paraformaldehyde for 15 minutes (at 37°C) and permeabilized 30 min in PBS containing 10% FCS and 0.1% saponin or briefly extracted with an extraction buffer (0.5% TX-100, 10% glycerol in EMT [60 mM PIPES-NaOH pH6.9, 25 mM HEPES, 10 mM EGTA, 10 mM $MgCl_2$]) to obtain cytoskeletons and fixed in 3% paraformaldehyde in PBS for 15 minutes (at 37°C). Samples were then processed for immunofluorescence as in [9]. The primary antibodies (anti-BILBO1 1–110, 1:4000 dilution; anti-living colours rabbit polyclonal Clontech, 1:1,000 dilution; anti-HA tag mouse monoclonal IgG1 Biolegend, 1:1,000 dilution) were incubated for 1h in a dark moist chamber. After two PBS washes, cells were incubated for 1h with the secondary antibodies anti-rabbit IgG conjugated to Alexa fluor 594 (Molecular Probes, 1:400); anti-mouse IgG conjugated to FITC (Sigma, 1:400). The nuclei were stained with DAPI (0.20 μg.mL⁻¹ in PBS for 5 min), then washed twice in PBS and mounted overnight with Prolong (Molecular Probes P-36934). Images were acquired on a Zeiss Imager Z1 microscope with Zeiss 100x or 63x objectives (NA 1.4), using a Photometrics Coolsnap HQ2 camera and Metamorph software (Molecular Devices), and processed with ImageJ.

**Ultrastructure Expansion Microscopy.** The protocol was adapted from [23,35]. *T. brucei* PCF cells expressing endogenously ₁₀myc BILBO2 and ₁₀TY1 FPC4 (500 μl of culture corresponding to 4.10⁶ cells per coverslip) were loaded on poly-L-lysine coated 12-mm coverslips in 24-well plate and cells left to adhere between 5 to 10 minutes and extracted as described above. Cytoskeletons were covered with 1 mL of activation solution (0.7% formaldehyde; 1% Acrylamide in PBS) for 4 hours at 37°C with slow agitation. For the gelation step, coverslips were gently deposited on top of a 35 μl drop of MS solution (23% Sodium acrylate (Sigma 408220); 10% Acrylamide (Euromedex EU0060-A); 0.1% Bis-acrylamide (Euromedex EU0560-A) in

PBS) for 2 min then transferred to a 35 μl drop of MS; 0.5% TEMED (Euromedex 50406-A); 0.5% Ammonium persulfate (Euromedex EU0009-A) for 5 minutes on ice then transferred at 37˚C and incubated for 1 hour in a moist chamber without agitation. The coverslips were then transferred in 6-well plate in 1 mL of denaturation solution (200 mM Sodium Dodecyl Sulfate; 200 mM Sodium chloride; 50 mM Tris pH 9.0) with agitation at RT for 15 min to detach the gel from the coverslip, then moved into a 1.5 ml Eppendorf centrifuge tube filled with denaturation solution and incubated at 95˚C for one 90 minutes. Gels were expanded in large volumes of deionized water (twice 30 min then overnight) then incubated in a large volume of PBS for 10 minutes (three times). Small pieces of the gels were processed for immuno-labelling as follows. The gels were preincubated in blocking solution (PBS, 2% BSA, 0.2% Tween-20) for 30 min (37˚C). The primary antibodies (rabbit anti-BILBO1 1–110, 1:1000 dilution; mouse IgG1 anti c-Myc clone 9E10, 1:500 dilution; mouse IgG2a anti c-Myc clone 9B11 (Cell Signalling), 1:1000 dilution; mouse IgG1 anti-TY1 (BB2) [34], 1:1000 dilution, mouse IgG1 anti-Tubulin (DM1A Sigma) 1:250; mouse IgG2a anti-TbSAXO mAb25 [36] 1:5) diluted in blocking solution were incubated for 3 hours in the dark at 37˚C with slow agitation. After three washes in blocking solution, gels were incubated with the secondary antibodies (anti-Rabbit Alexa fluor 594 conjugated (Molecular Probes, 1:500 dilution); anti-Mouse Alexa fluor 488 conjugated (Molecular Probes, 1:200 dilution; anti-Rabbit Alexa fluor 647 conjugated (Molecular Probes, 1:200 dilution); anti-Mouse IgG1 specific Alexa fluor 594 conjugated (Molecular Probes, 1:500 dilution); anti-Mouse IgG2a specific Alexa fluor 488 (Molecular Probes, 1:1000 dilution)) diluted in blocking solution for 3 hours in the dark at 37˚C with slow agitation. After three washes in blocking solution, gels were expanded a large volume of deionized water (twice 30 minutes then overnight). An expansion factor of 4.3 was determined using the ratio between the size of the coverslip (12mm) and the size of the gels after the first expansion.

Images were acquired on a Zeiss Imager Z1 microscope with Zeiss 63x oil objectives (NA 1.4), using a Photometrics Coolsnap HQ2 camera and Metamorph software (Molecular Devices), and processed with ImageJ. U-ExM images were acquired on a Zeiss Imager Z1 microscope (Figs 6C and S6A) and a Leica SP8 WLL2 on an inverted stand DMI6000 (Leica Microsystems, Mannheim, Germany), using a 63X oil objective (HCX Plan Apo CS2, NA 1.40) (S6B Fig and S1 Movie) and processed with FIJI.

## Expression and purification of BILBO2-NTD and FPC4-B1BD

Recombinant BILBO2-NTD and FPC4-B1BD proteins were expressed in *E. coli* (strain BL21-DE3). Expression and purification of single proteins were carried out in a similar way to the co-expression described below. For co-expression, the two cloned constructs were co-transformed into competent bacterial cells. The cells were grown in Luria-Bertani (LB) medium at 37˚C to an $OD_{600}$ of ~0.6 and then subjected to cold shock on ice for 20 min. Protein expression was induced by addition of 0.5 mM isopropyl β-D-thiogalactopyranoside, and cell cultures were further incubated at 16˚C overnight (~16 h). Cells were harvested by centrifugation in a Sorvall GS3 rotor (6,000 × g, 12 min, 4˚C) and then resuspended in the lysis buffer containing 20 mM Tris-HCl, pH 8.0, 100 mM NaCl, 20 mM imidazole, 10 mM β- mercaptoethanol, and 5% (v/v) glycerol (20 ml buffer per litre of cell culture). Cells were lysed in an Emulsiflex C3 homogenizer (Avestin) and cell debris was pelleted by centrifugation (40,000 × g, 30 min, 4˚C). The supernatant was filtered (0.45-μm pore size, Amicon) and loaded onto a Ni-HiTrap column (GE Healthcare) which was pre-equilibrated with the same lysis buffer. The column was washed with 5 × column volume (CV) of lysis buffer, and bound protein was eluted using a linear gradient concentration of imidazole (20–500 mM, 20 × CV).

The His$_6$-SUMO tag on both proteins was of TbBILBO1-NTD was cleaved off by incubating the eluted proteins with ~1% (w/w) of SENP2 (4°C, overnight). Target proteins were further purified on a 16/60 Superdex-200 column (GE Healthcare) pre-equilibrated with a running buffer containing 20mM Tris-HCl (pH 8.0) and 100 mM NaCl. Elution peaks containing both proteins were pooled and concentrated to ~20 mg/ml for crystallization trials.

## Crystallization and structure determination

Purified protein of the BILBO2-NTD/FPC4-B1BD complex was used to set up crystallization trials. Initial conditions giving small seed-like crystals were further optimized. Single diamond-like crystals with dimensions of approximately $100 \times 50 \times 50$ μm were obtained under a condition containing 0.1 mM NaOAc (pH 4.5), 0.5 M 1,6-Hexanediol, and 10 mM CoCl$_2$.

Crystals were harvested by sequentially soaked in the same crystallization solution with increasing concentration of glycerol [5–20% (v/v)], mounted onto nylon loops, and then flash-frozen in liquid nitrogen. Data collection was carried out at the beamline ID23-1 of the European Synchrotron Radiation Facility (ESRF) at the wavelength of 0.9763 Å. Diffraction data were processed by the XDS program [37]. Structure determination was carried out using the molecular replacement method by the program Phaser within the Phenix suite [38]. Partially built models were checked and missing loops manually added in COOT [39]. Structure refinement was carried out using the phenix.refine [40].

## Yeast two-hybrid assay

Interactions assays were done on SC-W-L-H medium as described in the online protocol https://dx.doi.org/10.17504/protocols.io.btzenp3e. Photos were acquired after 3 days of incubation.

## Western-blotting

**Sample preparation for whole cells and cytoskeleton.** $2.10^7$ cells were split for whole cells (WC) and cytoskeleton (CSK) samples. For WC samples, cells were spun at 1,000 x g for 10 minutes, washed once and resuspended at $1.10^6$ cells/μL in PBS. An equivalent volume of 2x sample buffer and 25U of benzonase (Sigma, E1014) was added before boiling 5 minutes. For CSK samples, cells were spun at 1,000 g for 10 minutes and washed once in PBS, EDTA 10mM and resuspended at $1.10^6$ cells/μL$^{-1}$ in 100 mM PIPES pH6.9, 2 mM MgCl$_2$, 0.25% NP-40 (Igepal Sigma), Protease inhibitor (Calbiochem, 1:10,000 dilution) and 25U of benzonase. After 10 minutes incubation on ice, cytoskeletons were pelleted at 1,000 x g for 30 minutes then washed in 1 mL 100 mM PIPES pH6.9, 2 mM MgCl$_2$ and resuspended in the same buffer ($1.10^6$ cells/μL final). An equivalent volume of 2x sample buffer was added before boiling for 5 minutes. Protein samples (equivalent to $5.10^6$ trypanosomes whole cell or cytoskeletons) were separated on SDS-PAGE gels and transferred by semi-dry (BioRad) blotting 45 min at 25V on PVDF or PVDF low fluorescence membrane. After a 1 h blocking step in 5% skim milk in TBS-0.2% Tween-20, the membranes were incubated overnight at 4°C with the primary antibodies diluted in blocking buffer (anti-enolase (rabbit polyclonal, 1:25,000) [41]; anti-tubulin TAT1 (mouse monoclonal, 1:1,000 [42]); anti-BILBO1 1–110 (rabbit polyclonal, 1:1,000 [15]); anti-TY1 BB2 (mouse monoclonal 1:50,000 [34]); anti-HA (Biolegend mouse monoclonal, 1:1,000); anti-BILBO2 (guinea pig polyclonal, 1:100); anti-myc (mouse monoclonal 9E10, 1:10,000)). After three washes in blocking buffer, the membranes were incubated with the secondary antibodies (anti-mouse HRP-conjugated (Jackson Immunoresearch, 1:10,000); anti-rabbit HRP-conjugated (Sigma, 1:10,000 dilution); anti-guinea pig HRP-conjugated (Jackson Immunoresearch, 1:10,000); anti-mouse StarBright Blue 520 conjugated (Bio-Rad 12005867,

1:2,500); anti-rabbit StarBright Blue 700 conjugated (Bio-Rad 12004162, 1:2,500)) and washed twice 10 min in blocking buffer and twice 5 min in PBS. Blots were revealed using the Clarity Western ECL Substrate kit (Bio-Rad) with the ImageQuant LAS4000 (GE Healthcare) and with Chemidoc MP (Bio-Rad) for fluorescence.

## Isothermal titration calorimetry (ITC)

Purified BILBO2-NTD and FPC4-B1BD (WT or mutated) were dialyzed overnight against a buffer containing 20 mM Tris-HCl (pH 8.0) and 50 mM NaCl. Protein concentration was determined by ND-1000 spectrophotometer (PEQlab). ITC experiments were carried out at 25˚C using an iTC200 microcalorimeter (MicroCal, GE healthcare). The cell contained 200 μl of 50 μM BILBO2-NTD constructs, which was titrated with an initial 0.4 μl injection followed by 19 constitutive injections (2 μl each) of 600 μM FPC4-B1BD with a duration of 0.8 s. The interval between every two injections was 150 s. The ITC data were analysed using the program Origin version 7.0 provided by MicroCal. The One-site binding model was used to fit the integrated data to calculate the stoichiometry and binding constants.

## Supporting information

**S1 Fig. BILBO2 is cytosolic in U-2 OS cells. Tagged $_{HA}$BILBO2-T1, $_{HA}$BILBO2-T2, and $_{HA}$BILBO2-B1BD were expressed in U-2 OS cells and immunolabelled using anti-HA and anti-BILBO1 on fixed whole cells. Scale bars, 10 μm.**
(TIF)

**S2 Fig. Cellular localization of BILBO2 during the *T. brucei* BSF cell cycle.** A. Growth curves of PCF and BSF expressing endogenous tagged $_{TY1}$BILBO2. B. Co-immunolabelling of BILBO1 and $_{TY1}$BILBO2 on detergent-extracted cells using anti-BILBO1 and anti-TY1 antibodies. Scale bars, 5 μm.
(TIF)

**S3 Fig.** A. Growth curves of WT PCF cells, and non-induced and induced cells for ectopic expression of BILBO2$_{HA}$, BILBO2-T1$_{HA}$, BILBO2-T2$_{HA}$, and BILBO2-B1BD$_{HA}$. B. Western blot analysis of the expression level of BILBO2 and BILBO2$_{HA}$. Whole cell (WC) or detergent-extracted cytoskeleton (CSK) were labelled using the anti-BILBO2 polyclonal antibody (that recognises WT BILBO2 and induced BILBO2HA) and the anti-HA tag antibody that recognises induced BILBO2$_{HA}$ only. Loading and detergent-extraction controls were anti-BILBO1 and anti-Enolase. C. Western blot analysis of the level of expression of BILBO2$_{HA}$ and domains in WC and CSK. Loading and detergent-extraction controls were anti-PFR2 and Enolase. In A, B and C, cells were induced for 24H with 1 μg.mL$^{-1}$ tetracycline.
(TIF)

**S4 Fig. BILBO2 RNAi knock-down affects neither PCF nor BSF cell growth. Growth curves and western blotting analysis of *BILBO2* RNAi in $_{TY1}$BILBO2 expressing PCF and BSF cells. The anti-enolase was used as a loading control.**
(TIF)

**S5 Fig.** Growth curves of WT PCF cells, and non-induced and induced cells for ectopic expression of BILBO1$_{HA}$ (A) and BILBO2$_{HA}$ (B) and chimeric BILBO2-BILBO1$_{HA}$ (C) and BILBO1-BILBO2$_{HA}$ (D).
(TIF)

**S6 Fig.** A. Epifluorescence image of U-ExM triple labelling of tubulin, BILBO1 and TbSAXO, an axonemal protein. The arrowheads indicate the MTQ. Scale bar, 20 μm. B. 3D rendering of

confocal analysis of U-ExM co-labelling of BILBO1 and $_{myc}$BILBO2 that were used to generate S1 Movie.
(TIF)

**S7 Fig. Interaction between BILBO2-NTD and FPC4-B1BD.** A. Zoom-in view of the central part of the interface between BILBO2 and FPC4 with the $2F_o$-$F_c$ map (grey) contoured at 1.5 σ level. An ordered water molecule form multiple hydrogen bonds with residues from both proteins. B. Details of the interaction network between BILBO2 and FPC4. The plot was generated using DIMPLOT in the LigPlot plus suite.
(TIF)

**S8 Fig. Fate of BILBO2 and FPC4 in RNAi knockdown cell lines.** A. Growth curve of SmOxP427 cells expressing $_{myc}$BILBO2 and $_{TY1}$FPC4, non-induced and induced for *FPC4* RNAi or for *BILBO2* RNAi. B. Western blot analysis of whole cell (WC) and detergent-extracted cytoskeleton (CSK) SmOxP427 cells expressing $_{TY1}$FPC4 and $_{myc}$BILBO2 and induced 24-72h for *FPC4* RNAi or for *BILBO2* RNAi. C. Immunofluorescence labeling of BILBO1, $_{TY1}$FPC4, and $_{myc}$BILBO2 on whole cells 72h-induced for *FPC4* RNAi or for *BILBO2* RNAi. Note: no cytosolic pool was observed for FPC4 and for BILBO2 in the induced cells. Scale bars, 5 μm and 1 μm in insets.
(TIF)

**S1 Movie. Movie of 3D rendering after confocal analysis of BILBO1 (magenta) and $_{myc}$-BILBO2 (yellow) co-labelling.**
(AVI)

**S1 Data. Quantification raw data.** Excel spreadsheet containing, in separate sheets, the raw data for the immunofluorescence quantification in Fig 3D and western blot quantification in Fig 4D.
(XLSX)

## Acknowledgments

We thank Annelise Sahin and Marie Eggenspieler for their help at the beginning of the study.

We thank F. Bringaud (University of Bordeaux) for the anti-enolase antibody, K. Gull (Oxford University) for the anti-tubulin TAT1 antibody, P. Bastin (Institut Pasteur) for the anti-Ty1 antibody, Samuel Dean (Warwick Medical School) and Jack Sunter (Oxford Brookes University) for the pPOT plasmids and SmOx *T. brucei* cell lines. We are grateful to the staff at the beamline of ID23-1 at the European Synchrotron Radiation Facility (ESRF) for their help with X-ray diffraction. The help of Magali Mondin (Bordeaux Imaging center) is acknowledged. We thank J. Marcos, G. Cougnet-Houlery, and S. Guit for the continued MFP lab infrastructure. We thank Dr A. Albisetti for critically reading the manuscript.

## Author Contributions

**Conceptualization:** Gang Dong, Mélanie Bonhivers.

**Data curation:** Gang Dong, Mélanie Bonhivers.

**Formal analysis:** Paul Majneri, Yulia Pivovarova, Johannes Lesigang, Gang Dong, Mélanie Bonhivers.

**Funding acquisition:** Derrick R. Robinson, Gang Dong, Mélanie Bonhivers.

**Investigation:** Derrick R. Robinson, Gang Dong, Mélanie Bonhivers.

**Methodology:** Charlotte Isch, Paul Majneri, Nicolas Landrein, Yulia Pivovarova, Johannes Lesigang, Florian Lauruol, Gang Dong, Mélanie Bonhivers.

**Project administration:** Gang Dong, Mélanie Bonhivers.

**Resources:** Gang Dong, Mélanie Bonhivers.

**Supervision:** Gang Dong, Mélanie Bonhivers.

**Validation:** Gang Dong, Mélanie Bonhivers.

**Visualization:** Gang Dong, Mélanie Bonhivers.

**Writing – original draft:** Gang Dong, Mélanie Bonhivers.

**Writing – review & editing:** Derrick R. Robinson, Gang Dong, Mélanie Bonhivers.

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
