## [Decision Letter · Decision Letter 0]

23 Feb 2021

Dear Dr. BONHIVERS,

Thank you very much for submitting your manuscript "Structural and functional studies of the first tripartite protein complex at the Trypanosoma brucei flagellar pocket collar" for consideration at PLOS Pathogens. As with all papers reviewed by the journal, your manuscript was reviewed by members of the editorial board and by several independent reviewers. In light of the reviews (below this email), we would like to invite the resubmission of a significantly-revised version that takes into account the reviewers' comments.

This manuscript was assessed by 3 reviewers, while all were positive about this manuscript, reviewer 2 wanted to see two additional experiments:

i) A conditional KO to investigate further whether BILBO2 is truly essential as the inability to generate a KO suggests.

ii) Further investigation of the tripartite dependencies through additional combinations of RNAi and localisation.

Given the overall reviewers positive response to this manuscript and the extensive other data within the paper we feel that i). is not necessary. However, ii). testing reciprocal FPC4 / BILBO2 localisation dependencies and hierarchies is a potentially informative set of additional experiments and given the reagents/cell lines available this is achievable in the revision timescale so you should address this.

In addition, all reviewers had a number of other points requiring textual changes that should be addressed.

We cannot make any decision about publication until we have seen the revised manuscript and your response to the reviewers' comments. Your revised manuscript is also likely to be sent to reviewers for further evaluation.

Sincerely,

Jack Sunter

Guest Editor

PLOS Pathogens

David Horn

Section Editor

PLOS Pathogens

Kasturi Haldar

Editor-in-Chief

PLOS Pathogens

orcid.org/0000-0001-5065-158X

Michael Malim

Editor-in-Chief

PLOS Pathogens

orcid.org/0000-0002-7699-2064

This manuscript was assessed by 3 reviewers, while all were positive about this manuscript, reviewer 2 wanted to see two additional experiments:

i) A conditional KO to investigate further whether BILBO2 is truly essential as the inability to generate a KO suggests.

ii) Further investigation of the tripartite dependencies through additional combinations of RNAi and localisation.

I feel the demonstration of BILBO2 essentiality or not will not provide additional insight and given the overall reviewers positive response to this manuscript and the extensive other data within the paper this experiment is not necessary. However, testing reciprocal FPC4 / BILBO2 localisation dependencies and hierarchies is a potentially informative set of additional experiments and given the reagents/cell lines available this is achievable in the revision timescale so you should address this.

In addition, all reviewers had a number of minor issues requiring textual changes that should be addressed.

Reviewer's Responses to Questions

**Part I - Summary**

Reviewer #1: The T. brucei FPC is a vital structural component of the trypanosome cell. Despite the importance of the FPC, prior to this study, only four proteins of this structure had been identified and two characterised. This study identifies a further FPC protein, BILBO2, and convincingly demonstrates its capacity to interact with both BILBO1 and FPC4. The study also includes a crystal structure that reveals the mode of binding of BILBO2 to FPC4 and identifies the key residues involved. Together these data are of strong quality, important, novel and publishable. However, the manuscript also indicates that a molecular mechanism by which BILBO2 it is able to influence the BILBO1 filament shape has been elucidated and, in my opinion, the data does not support that conclusion.

Supported claims

1) BILBO2 is a BILBO1 interacting protein, dependent on a B1BD.

The interaction is supported in two different heterologous systems; Y2H assays (Fig 1) and in U-2 OS mammalian cells (Fig 2) with the same B1BD of the BILBO2 protein necessary and sufficient for the interaction in both systems. On detergent extracted cells, the two proteins also co-localise throughout the cell cycle in T. brucei, again in a B1BD-dependent manner (Fig 3, S6, S7). This claim is therefore, well supported by the data in the manuscript.

2) BILBO2 interacts with FPC4

The interaction is supported with Y2H assays, in U-2 OS mammalian cells and through partial co-localisation of BILBO2 and FPC4 (all Fig 6). Again, domains required for interactions are consistent between methods. This interaction, and the domains responsible, are fully validated with ITC (Fig 8) and a crystal structure (Fig 7). Two residues predicted to be important for this interaction, based on the crystal structure, were additionally shown to affect protein binding by ICT and in vivo (Fig 8). This claim is therefore strongly supported by the evidence provided.

Reviewer #2: African trypanosomes are kinetoplastid parasites that cause the disease African Sleeping Sickness. The flagellar pocket is central to their pathogenicity as it is the only site of endocytosis and it sequesters the cell’s surface receptors, meaning that the FP is the hub of host-parasite interaction. The Flagellar pocket collar (FPC) is essential for FP biogenesis and interfering with its function causes parasite cell death.

In this study, the authors characterise one of the hits (BILBO2) from a previous Y2H screen on BILBO1, the major structural component of the trypanosome flagellar pocket collar (FPC). They use a variety of in vivo expression systems (yeast, mammalian cells and trypanosomes) to demonstrate direct interaction between specific domains of BILBO1, BLBO2 and FPC4 (termed “tripartite” by the authors here). They raise antibodies and use epitope tagging to demonstrate localisation of BILBO2 at the FPC, and also use 3-colour expansion microscopy to demonstrate almost perfect colocalisation of BILBO1 and BILBO2, with close localisation of FPC4. The authors use X-ray crystallography to solve the structure o the BILBO2-NTD/FPC4-CTD complex.

The data is high quality, the experiments innovative and well executed and, with some exceptions detailed below, the conclusions are warranted from the data. The major concern I have with the study is the absence of functional insight. No data is presented to indicate BILBO2 function, provide additional insights into FPC assembly, or that the tri-partite complex is important in FPC assembly. Some functional insights are warranted for this work to be published in PLIOS Pathogens.

Reviewer #3: Using a combination of techniques the authors identified a novel component of the flagellar pocket collar, a curious structure that is required to form the flagellar pocket, which is an membranous invagination at the base of the single flagellum of the human and animal parasite Trypanosoma brucei. The authors very clearly demonstrate the overlapping localization with a previously described flagellar pocket collar protein (Bilbo 1) that forms a ring like structure in the FPC region. They demonstrate this in situ as well as in a human cell system where these two proteins form large filamentous complexes. They go on to map the interacting domains of these two proteins and with FPC4 another protein of this structure. They furthermore show that Bilbo 2 requires Bilbo 1 for proper localization and that Bilbo 2 likely functions in shape determination of the Bilbo 1 filaments. Finally they solve the structure of the Bilbo1/2 complex and demonstrate the mechanism of interaction. The manuscript is well written, figures are clear and mostly with good figure legends. The data is of very high quality (I can not judge the structure data I should add). I wish I were an author on this manuscript. For the future I strongly recommend to put this amount of work in at least two PLOS Pathogens manuscripts!!

The manuscript should be accepted with minor adjustments to text and figure legends.

**Part II – Major Issues: Key Experiments Required for Acceptance**

Reviewer #1: None

Reviewer #2: The authors propose that the lack of RNAi phenotype is due to incomplete penetrance, and use Alsford et al RITseq data and their inability to generate a KO to suggest BILBO2 is in fact essential. However, the lack of phenotype could be because BILBO2 is not essential for FPC biogenesis (or for any other in vitro cellular function), a RITseq phenotype could be a false positive in this dataset, and the inability to generate a KO could be due to technical challenges. To clarify this issue, the authors should make a BILBO2 KO with an inducible, ectopic addback and perform an analysis on any resulting phenotype.

The authors propose a tri-partite protein complex at the FPC, and ablate BILBO1 to demonstrate BILBO2’s dependency upon BILBO1. However, this is a rather extreme test of dependency because the FPC is not present in BILBO1 RNAi cells, and is therefore not a very informative experiment. To gain further insights into BILBO1-BILBO2-FPC4 tri-partite dependencies and hierarchies, the authors should ablate FPC4, BILBO2 and examine the affect upon BILBO2 and FPC4 localisation. It would be informative (but not absolutely necessary) to determine whether BILBO2 and FPC4 interact in the cytosol of BILBO1 ablated cells.

Reviewer #3: (No Response)

**Part III – Minor Issues: Editorial and Data Presentation Modifications**

Reviewer #1: The authors state on page 8 that “these data identify BILBO2 as a novel BILBO1-binding protein that plays a role in the oligomerization of BILBO1” and in the abstract “BILBO2 colocalizes with BILBO1 and can modulate the shape of the BILBO1 filament by interacting with the BILBO1 EF-hand domains”. The data does not convincingly demonstrate that BILBO2 plays a role in oligomerisation of BILBO1 and therefore, although this possibility should be discussed in the discussion, it should not be presented as a finding unless the authors would like to add additional data to support this. While BILBO2 does seem to facilitate the production of short filaments in the mEFh1-expressing cells, WT BILBO1 is able to oligomerise without BILBO2 (Fig 2Aa), whereas BILBO2 does not oligomerise without BILBO1 (Fig 2Ab / Fig S1) or localise to the FPC (Fig 4). Indeed, from the data in this paper, BILBO2 appears to require BILBO1 or FPC4 for its localisation and/or filament formation and not the other way around.

Other minor comments:

Figure 3 is focussed on the detergent extracted cells for co-localisation of BILBO1 and BILBO2 in these cells. For panel C the whole cells clearly have BILBO2 distributed throughout the cell as well as the FPC. This is not discussed in the text, but presumably, given the data in Fig 4, this is due to overexpression of the ectopically-expressed BILBO2. That should be stated in the text (ideally along with an accompanying western blot to demonstrate extent of overexpression).

Figure 3 legend has panel C and D descriptions the wrong way round.

Figure 4B: 48 hour cell should include inset panel (as for the NI and 24 hour cells) to support statement on page 10 that “…after 48h of induction BILBO2 was neither detected at the old flagellum nor at the new flagellum”.

No scale bar on inset panels of Figures 4B or 5B.

Resolution of figures in reviewers PDF are lower than publication quality and hard to read small text, especially Fig 6.

Material spelt incorrectly on page 18.

*Line numbers should have been included for the review process.*

Reviewer #2: Figure 3 quantitation suggests that BILOBO2 protein intensity is half that of BILBO1 in late cell cycle stages, but this is not apparent on the example micrographs. Presumably this is an issue of contrast settings - it would be helpful for the reader to have the same contrast settings on early and late cells so the reduction in signal intensity can be observed.

More details should be provided (in the text of Figure legend) on what is meant by BILBO1 and 2 fluorescence intensity at the old and new FPC. i.e. is this sum intensity per collar, or average intensity per pixel?

Figure 3D and 3C are mixed up in the Figure legend.

In figure 3, the data and text on over-expression of BILBO2-T2 in trypanosomes is confusing and contradictory. The text states that BILBO2-T2 localises to the FPC in whole cells but is mostly extracted by detergent, concluding that this mutant has FPC localisation but is not fully functional. However, I am unable to see this mutant localise to the FPC on whole cells. In contrast, this mutant DOES localise clearly and brightly to the FPC in cytoskeletons, but only in the enlarged inset, and it is not evident at all in the main panel. This discrepancy in the text and data need to be resolved.

Figure 4C – typo (Tubuline should be Tubulin)

Figure 5A legend states that asterisks indicate identical residues – this is not true in several cases when examining the alignment.

Figure 5 legend typo – columns should be colons

Figure 5 B legend – amend to state these are cytoskeletons, rather than whole cells

Figure 5B and associated text: The authors convincingly demonstrate that the BILBO1/2 NTD chimeras localise to the FPC in cytoskeletons, although their experiment would have been more compelling if they had used anti-Ty labelling of Ty::BILBO1 to prevent anti-BILBO1 cross-reactivity with the chimera in their colocalisation fluorescence microscopy. The authors previously show (Figure 3C) that B1BD is necessary and sufficient for FPC localisation, and the localisation of BILBO1 to the FPC has not been shown to require its NTD. Therefore, there is nothing to suggest that the localisation of these mutant proteins to the FPC is due to the presence of the chimeric BILBO1/2 NTDs. Moreover, the authors state their data supports the hypothesis that the BILBO1/2 NTDs share a similar function. However, they only present localisation data, not functional data, so no functional inferences can be drawn. For functional insights, the authors must demonstrate that the chimeras can perform the function of the wildtype protein (e.g. rescue of an RNAi phenotype).

Figure 6A: regarding the second Y2H in the series – could the authors confirm whether this was intended to indicate be a 3-way Y2H, between BILBO2-NTD, BILBO2-B1BD, and FPC4-B1BD?

Reviewer #3: Minor changes in the figures/figure legends:

Figure 1 B: I am not sure I understand the fourth row where B1BD is tested as prey and bait? Also the -Control and +Control are not well explained in the figure legends. Also please stay consistent with the naming scheme Bilblo1-BD and B1BD is the same I assume no need to have different names then.

Figure 3:

- C and D are reversed in comparison to their description in the figure legend

- In the figure legend part C, please mention the number of cells this has been quantified with

Figure 6:

- In figure 6 C, please double-check the scale bars. It is decribed to be 5um, but with expansion, the cells should be much bigger than that I expect

Minor changes in the text:

- Page 2 text passage “…demonstrate that BILBO1 and BILBO2 share a homologous NTD domain and that both domains…”: delete the word domain. That word is already part of the abbrievation NTD.

- Page 9/10 the passage about the BILBO2 knockdown. The authors explain that RNAi of BILBO2 had no effect, and they give the remaining trace amounts of BILBO2 as a possible explanation for this behaviour. In their data, they give some evidence that to form a flagellar pocket, the parasite indeed doesn’t need 100% of the BILBO2 a G1 cell has (Figure 3D). They show that a signal of 50% per flagellar pocket collar in 1K1N-2FPC, as observed in IFA, is enough for the pocket to build. This supports the idea that an incomplete knockdown would not be enough to efficiently disrupt flagellar pocket collar formation. Thus, the assumption of the authors that the incomplete depletion of BILBO2 is the reason for the missing phenotypical effect, is true. But they should make use of their own data to support their conclusion.

- Page 12 teyt passage “…probably due to primary and secondary antibodies steric hindrance, with turned to…”: Typo, with = we

Please do not refer to other publications for the methods. Describe the methods in the manuscript and add references where appropriate (in particular for the Yeast two-hybrid assay).

Please Adjust the reference for Amodeo et al. 2020 to the now published version: Journal of Cell Science 2021 : jcs.254300 doi: 10.1242/jcs.254300 Published 15 February 2021

PLOS authors have the option to publish the peer review history of their article (what does this mean?). If published, this will include your full peer review and any attached files.

Reviewer #1: No

Reviewer #2: No

Reviewer #3: No
---

## [Editor Report · Decision Letter 1]

28 Jun 2021

Dear Dr. BONHIVERS,

Thank you very much for submitting your revised manuscript "Structural and functional studies of the first tripartite protein complex at the Trypanosoma brucei flagellar pocket collar" for consideration at PLOS Pathogens. We are likely to accept this manuscript for publication, but ask that you address the points below and modify the manuscript accordingly.

The abstract needs to be reworked to reflect the comments by reviewer 1 on the effect of BILBO2 on BILBO1 polymers. Currently, it states "... and can modulate the shape of the BILBO1 filament by interacting with the BILBO1 EF-hand domains." which appears to be too strong a conclusion from the evidence presented.

Plus on line 463 it should be unknown.

Sincerely,

Jack Sunter

Guest Editor

PLOS Pathogens

David Horn

Section Editor

PLOS Pathogens

Kasturi Haldar

Editor-in-Chief

PLOS Pathogens

orcid.org/0000-0001-5065-158X

Michael Malim

Editor-in-Chief

PLOS Pathogens

orcid.org/0000-0002-7699-2064

Figure Files:

Data Requirements:

Reproducibility:

References:

---

## [Editor Report · Decision Letter 2]

4 Jul 2021

Dear Dr. BONHIVERS,

We are pleased to inform you that your manuscript 'Structural and functional studies of the first tripartite protein complex at the Trypanosoma brucei flagellar pocket collar' has been provisionally accepted for publication in PLOS Pathogens.

Best regards,

Jack Sunter

Guest Editor

PLOS Pathogens

David Horn

Section Editor

PLOS Pathogens

Kasturi Haldar

Editor-in-Chief

PLOS Pathogens

orcid.org/0000-0001-5065-158X

Michael Malim

Editor-in-Chief

PLOS Pathogens

orcid.org/0000-0002-7699-2064
---

## [Editor Report · Acceptance letter]

28 Jul 2021

Dear Dr. Bonhivers,

We are delighted to inform you that your manuscript, "Structural and functional studies of the first tripartite protein complex at the Trypanosoma brucei flagellar pocket collar," has been formally accepted for publication in PLOS Pathogens.

Best regards,

Kasturi Haldar

Editor-in-Chief

PLOS Pathogens

orcid.org/0000-0001-5065-158X

Michael Malim

Editor-in-Chief

PLOS Pathogens

orcid.org/0000-0002-7699-2064